# Generation of a mutator parasite to drive resistome discovery in *Plasmodium falciparum*

Krittikorn Kümpornsin[1,2], Theerarat Kochakarn[3], Tomas Yeo [4], John Okombo [4], Madeline R. Luth[5], Johanna Hoshizaki [1], Mukul Rawat[1], Richard D. Pearson [1], Kyra A. Schindler [4], Sachel Mok [4,6], Heekuk Park [7], Anne-Catrin Uhlemann [6,7], Gouranga P. Jana[8], Bikash C. Maity[8], Benoît Laleu [9], Elodie Chenu[9], James Duffy[9], Sonia Moliner Cubel[10], Virginia Franco[10], Maria G. Gomez-Lorenzo [10], Francisco Javier Gamo [10], Elizabeth A. Winzeler[5], David A. Fidock [4,6], Thanat Chookajorn[3,11] & Marcus C. S. Lee [1,12] ✉

In vitro evolution of drug resistance is a powerful approach for identifying antimalarial targets, however, key obstacles to eliciting resistance are the parasite inoculum size and mutation rate. Here we sought to increase parasite genetic diversity to potentiate resistance selections by editing catalytic residues of *Plasmodium falciparum* DNA polymerase δ. Mutation accumulation assays reveal a ~5–8 fold elevation in the mutation rate, with an increase of 13–28 fold in drug-pressured lines. Upon challenge with the spiroindolone PfATP4-inhibitor KAE609, high-level resistance is obtained more rapidly and at lower inocula than wild-type parasites. Selections also yield mutants with resistance to an "irresistible" compound, MMV665794 that failed to yield resistance with other strains. We validate mutations in a previously uncharacterised gene, PF3D7_1359900, which we term quinoxaline resistance protein (QRP1), as causal for resistance to MMV665794 and a panel of quinoxaline analogues. The increased genetic repertoire available to this "mutator" parasite can be leveraged to drive *P. falciparum* resistome discovery.

Antimalarial drug discovery has been actively searching for new or improved medicines to treat and ultimately eliminate malaria. Current front-line artemisinin-based combination therapies (ACTs) for *Plasmodium falciparum* have been compromised by the emergence of parasites that are less susceptible to both artemisinin and partner drugs in Southeast Asia, an epicentre of antimalarial resistance[1,2]. Furthermore, artemisinin resistance is a public health threat to people living in endemic regions worldwide, as exemplified by recent reports

[1]Wellcome Sanger Institute, Wellcome Genome Campus, Hinxton, UK. [2]Calibr, Division of the Scripps Research Institute, La Jolla, CA, USA. [3]The Laboratory for Molecular Infection Medicine Sweden and Department of Molecular Biology, Umeå University, Umeå, Sweden. [4]Department of Microbiology and Immunology, Columbia University Irving Medical Center, New York, NY, USA. [5]Department of Pediatrics, School of Medicine, University of California, San Diego, La Jolla, CA, USA. [6]Center for Malaria Therapeutics and Antimicrobial Resistance, Division of Infectious Diseases, Department of Medicine, Columbia University Irving Medical Center, New York, NY, USA. [7]Division of Infectious Diseases, Department of Medicine, Columbia University Irving Medical Center, New York, NY, USA. [8]TCG Lifesciences Private Limited, Salt-lake Electronics Complex, Kolkata, India. [9]Medicines for Malaria Venture, International Centre Cointrin, Geneva, Switzerland. [10]Global Health Medicines R&D, GlaxoSmithKline, Tres Cantos, Madrid, Spain. [11]Genomics and Evolutionary Medicine Unit, Centre of Excellence in Malaria Research, Faculty of Tropical Medicine, Mahidol University, Bangkok, Thailand. [12]Biological Chemistry and Drug Discovery, Wellcome Centre for Anti-Infectives Research, University of Dundee, Dundee, UK. ✉e-mail: mlee001@dundee.ac.uk

of the emergence of *kelch13* mutations in Rwandan and Ugandan isolates that cause reduced susceptibility to artemisinin[3,4]. Many promising antimalarial compounds with good potency and multi-stage activity have been uncovered using whole-cell phenotypic-based screening[5]. However, this approach presents difficulties for lead optimisation because of the lack of knowledge of the molecular target. A deeper understanding of the drug target, mode of action, and resistance mechanism could lead to the design of better medicines that can withstand drug resistance[6–8]. In addition, knowledge of the drug target or resistance gene provide molecular markers for genomic epidemiology surveillance in the field to monitor the spread and containment of drug resistance[9,10].

In vitro evolution of drug resistance followed by whole-genome analysis has become a key approach for drug target identification by helping define modes of action as well as mechanisms and propensities for resistance[11–13]. A typical in vitro resistance selection is performed using parasite inocula ranging from $10^5$ to $10^9$ parasites, which are exposed to an antimalarial compound at a concentration capable of killing all the drug-sensitive parasites[8,14,15]. Recrudescent parasites can then be subjected to whole-genome sequencing to identify the underlying gene responsible for the resistance phenotype. The main obstacle to success is the prolongation or in some cases complete inability to select for resistant parasites, regardless of the selection regimen or strain background. This labour- and time-intensive process may thus fail to identify a molecular target or a defined mechanism of action for query compounds. For example, in a set of in vitro resistance selections with 48 compounds by the Malaria Drug Accelerator Consortium (MalDA), 23 compounds yielded resistant parasites with resistance observed after 15–300 days[16]. Compounds that fail to yield resistant parasites after multiple attempts and conditions have been termed "irresistible"[17]. Although there may be multiple possible reasons for compounds being apparently "irresistible", their low propensity for resistance is an attractive quality from an antimalarial drug discovery perspective and thus insights into their mechanism of action would be valuable.

The ability to select for a parasite with a protective mutation depends, at least in part, on an inoculum size with sufficient genetic variation. However, for reasons of technical practicality, the maximum inoculum for in vitro resistance selections is typically capped at ~$5 \times 10^9$ parasites per flask (~10% parasitemia with 3% hematocrit in a 170 mL culture), orders of magnitude less than what can occur in an infected human. Larger culture sizes and extended selection times also consume more compound, which may be limiting. Several laboratory strains, as well as field isolates collected from the drug resistance epicentre of western Cambodia, have been shown to have a similar range of mutation rates of around $10^{-9}$ to $10^{-10}$ base substitutions per site per asexual life cycle[18–21].

In this work, to increase the genetic diversity represented in a given culture volume and potentially shorten the experimental time scale of selection, we use CRISPR-Cas9 to generate a *P. falciparum* mutant Dd2 parasite that has deficient proof-reading activity of the DNA polymerase δ catalytic subunit, inspired by a similar approach in the rodent malaria parasite *Plasmodium berghei*[22,23]. We show that this engineered *P. falciparum* line has an increased mutation rate, lowering the inoculum and shortening the time required to select resistance to KAE609, a compound with a known target[24]. When challenged with a previously identified irresistible compound MMV665794 that had earlier failed in selections with wild-type 3D7 and Dd2 parasites[16,25], we obtain multiple resistant clones with mutations in a gene of unknown function, PF3D7_1359900. CRISPR-Cas9 editing of these candidate mutations into wild-type parasites confer a similar level of resistance to the selected line, demonstrating the role of this gene in resistance to quinoxaline-based compounds. Our results support the potential of this "mutator" parasite to identify new antimalarial targets and understand drug resistance mechanisms.

## Results

### CRISPR editing of DNA polymerase δ

To increase the genetic repertoire of *P. falciparum* parasites in culture, we aimed to generate parasites with impaired 3′–5′ proof-reading activity from the catalytic subunit of DNA polymerase δ (PF3D7_1017000) in order to increase the level of basal spontaneous mutations, based on prior work in yeast and the rodent malaria parasite *Plasmodium berghei*[22,23,26]. The high-fidelity replicative DNA polymerase δ is a major enzyme for lagging-strand synthesis and contains 3′–5′ exonuclease activity that can excise misincorporated nucleotides during DNA replication[27,28]. The disruption of two conserved catalytic residues of the exonuclease domain of DNA polymerase δ (Supplementary Fig. 1) leads to impaired 3′–5′ proof-reading activity, resulting in reduced fidelity in DNA replication. This causes an increase in nucleotide sequence variation and higher mutation rate in the genome[29,30]. The two conserved catalytic residues of the *P. falciparum* 3′–5′ proof-reading subunit, D308 and E310, were replaced with alanine using CRISPR-Cas9 in the Dd2 strain background (Fig. 1A, B).

*P. falciparum* DNA polymerase δ is predicted to be essential for parasite survival[31]. To examine whether ablation of the proof-reading function of DNA polymerase δ incurred a fitness cost to the parasite, we performed a competitive fitness assay. Dd2-GFP, an engineered parasite that strongly expresses green fluorescence protein (GFP), was used as a growth reference[32]. The reference Dd2-GFP line was mixed in a 1:1 ratio with either Dd2 wild-type (Dd2-WT) or the Dd2 DNA polymerase δ mutant (Dd2-Polδ), and the relative proportions of the two lines were measured by flow cytometry every two days for 20 days (~10 generations). Dd2-Polδ showed only a slightly reduced fitness compared with Dd2-WT based on how quickly each line was able to outcompete the more slowly proliferating Dd2-GFP reference (Fig. 1C).

### Impaired proof-reading DNA polymerase δ increases single nucleotide variants

To test for changes in nucleotide sequence diversity and mutation rate, we performed a mutation accumulation assay in combination with whole-genome sequencing (Fig. 2A), comparing Dd2-Polδ with Dd2-WT. Three clones of Dd2-Polδ (E8, F11, H11) and a clone of Dd2-WT were cultured in complete medium continuously for 100 days (~50 generations) (Fig. 2A). Parasites were collected every 20 days and clones isolated by limiting dilution. A total of 12 clones of Dd2-WT and 37 clones of Dd2-Polδ, corresponding to one to three clones per timepoint, were randomly selected for whole-genome sequencing. The genomes of all parasites were mapped to the Dd2 reference genome (PlasmoDB-44_PfalciparumDd2_Genome). The Dd2 core genome comprising coding and non-coding regions was employed as the reference for single nucleotide variant (SNV) calls. The variant surface antigen gene family (*var*) and subtelomeric region of all chromosomes were excluded from the core genome. The coordinates of the Dd2 core genome are defined in Supplementary Table 1. The de novo SNVs for each of the clones were determined by comparison with their parental lines on day 0. The number of de novo SNVs in Dd2-WT was on average less than 1 SNV per clone in the coding sequence over the 100-day culture period. In contrast, each of the Dd2-Polδ clones had on average 3–6 SNVs per clone in coding regions (exome) (Fig. 2B, Supplementary Fig. 2A, B, and Supplementary Table 2 and Supplementary Data 1). The difference in the number of SNVs in non-coding regions between Dd2-WT and Dd2-Polδ clones was less pronounced (Fig. 2B). Nonetheless, each of the Dd2-Polδ clones had a greater number of SNVs of all types, in particular non-synonymous variants, distributed across all 14 chromosomes (Fig. 2C and Supplementary Fig. 3A, B). We examined whether there were any clusters of mutations (arising within 20 bp), identifying 12 examples, however, these were all in AT-rich intergenic regions (Supplementary Table 3). We identified only four genes that contained multiple missense mutations

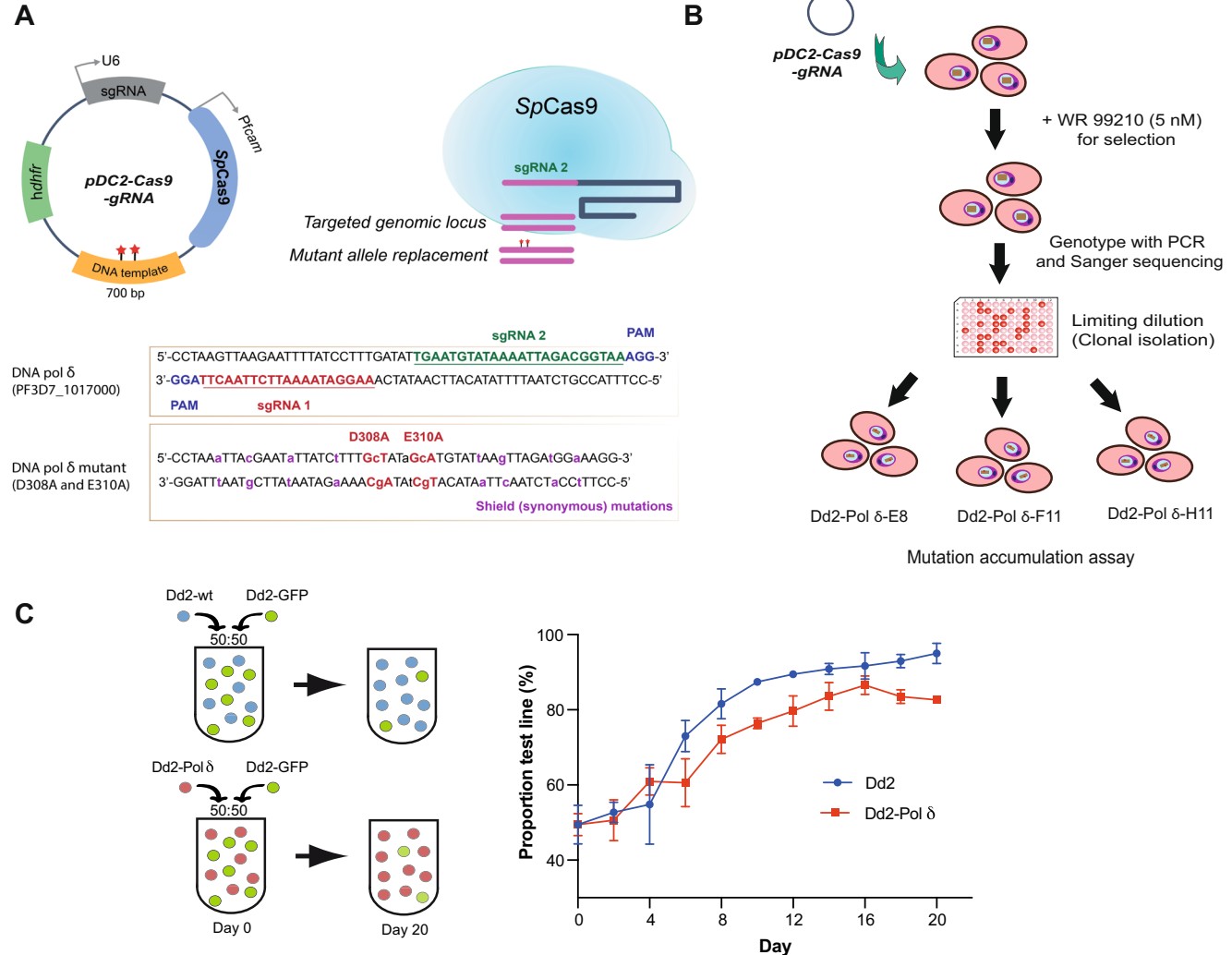

**Fig. 1 | CRISPR-Cas9 editing of the DNA polymerase δ proof-reading subunit.**
**A** The D308 and E310 residues were replaced by alanine in *P. falciparum* Dd2 by using the pDC2-Cas9-gRNA plasmid containing the sgRNA, Cas9 and donor template. The two sgRNA binding sites and the silent shield mutations are indicated. **B** CRISPR-Cas9 edited parasites were selected with 5 nM WR99210 and edited clones were isolated by limiting dilution. Three clonal DNA polymerase δ mutant parasites (Dd2-Polδ) were selected for whole-genome sequencing and the mutation accumulation assay. **C** Fitness of the Dd2-Polδ mutant parasite. Competitive fitness assays mixed the fluorescent reference line Dd2-GFP in a 1:1 ratio with either Dd2-WT or Dd2-Polδ clone H11. Growth was determined by flow cytometry every two days for a total of 20 days, with the proportion of GFP-positive parasites compared with total infected RBCs detected by MitoTracker Deep Red. Two independent biological experiments (*n* = 2) with technical triplicates were performed, data are presented as mean values +/−SD. Source data are provided as a Source Data file.

(Supplementary Fig. 3C). Comparison of base pair substitutions for transition (Ts) and transversion (Tv) events showed a moderate decrease in the Ts:Tv ratio in Dd2-Polδ and an increase in G:C → A:T transitions of 2-4 fold (Supplementary Fig. 4A, B).

We next determined the mutation rates for Dd2-WT and the three Dd2-Polδ clones E8, F11 and H11 based on the mean number of de novo SNVs for all clones of each line per generation, per genome or exome (see Methods). Each of the Dd2-Polδ clones showed higher mutation rates than Dd2-WT, varying from 2–3 fold in the core genome (coding and non-coding regions) to 5–8 fold in coding regions (exome) (Fig. 2D, Table 1 and Supplementary Data 2). Dd2-Polδ clones F11 and H11 showed a higher mutation rate than clone E8 (Fig. 2D and Table 1), and thus all subsequent experiments were performed with clone H11. To examine whether the modest differences in mutation rate between clones might be attributed to spontaneous mutations in DNA repair genes, we also looked at whether genes playing a role in DNA repair were mutated in the Dd2-Polδ lines during the 100-day culture period (Supplementary Data 1). Although SNVs within or near DNA repair genes were observed in each of the Dd2-Polδ lines, no single SNV was shared among all clones. Dd2-Polδ clone E8 possessed SNVs in the coding region of two putative DNA repair genes: a G435E change in DNA polymerase theta (PfDd2_130037000) and a N420K change in DNA repair protein RHP16 (PfDd2_120056000). Dd2-Polδ clone F11 had a P225L change in RuvB-like helicase 3 (PfDd2_130068000). Dd2-Polδ clone H11 did not have SNVs in the coding region of any DNA repair genes; however, a SNV was observed in the non-coding region in proximity to proliferating cell nuclear antigen 2 (PfDd2_120031600) (Supplementary Data 1).

## Dd2-Polδ potentiates in vitro drug resistance selections

Based on the assumption that a more diverse genetic repertoire available to the Dd2-Polδ cultures would increase the efficiency of selecting for drug-resistant parasites, we performed a proof-of-concept experiment comparing Dd2-WT with Dd2-Polδ using a drug with a well-characterised mode-of-action. KAE609 (cipargamin), currently in Phase II clinical trials, targets the *P. falciparum* P-type sodium ATPase 4 gene (*Pfatp*4, PF3D7_1211900) with SNVs known to confer resistance[24]. An in vitro drug resistance selection was performed with a

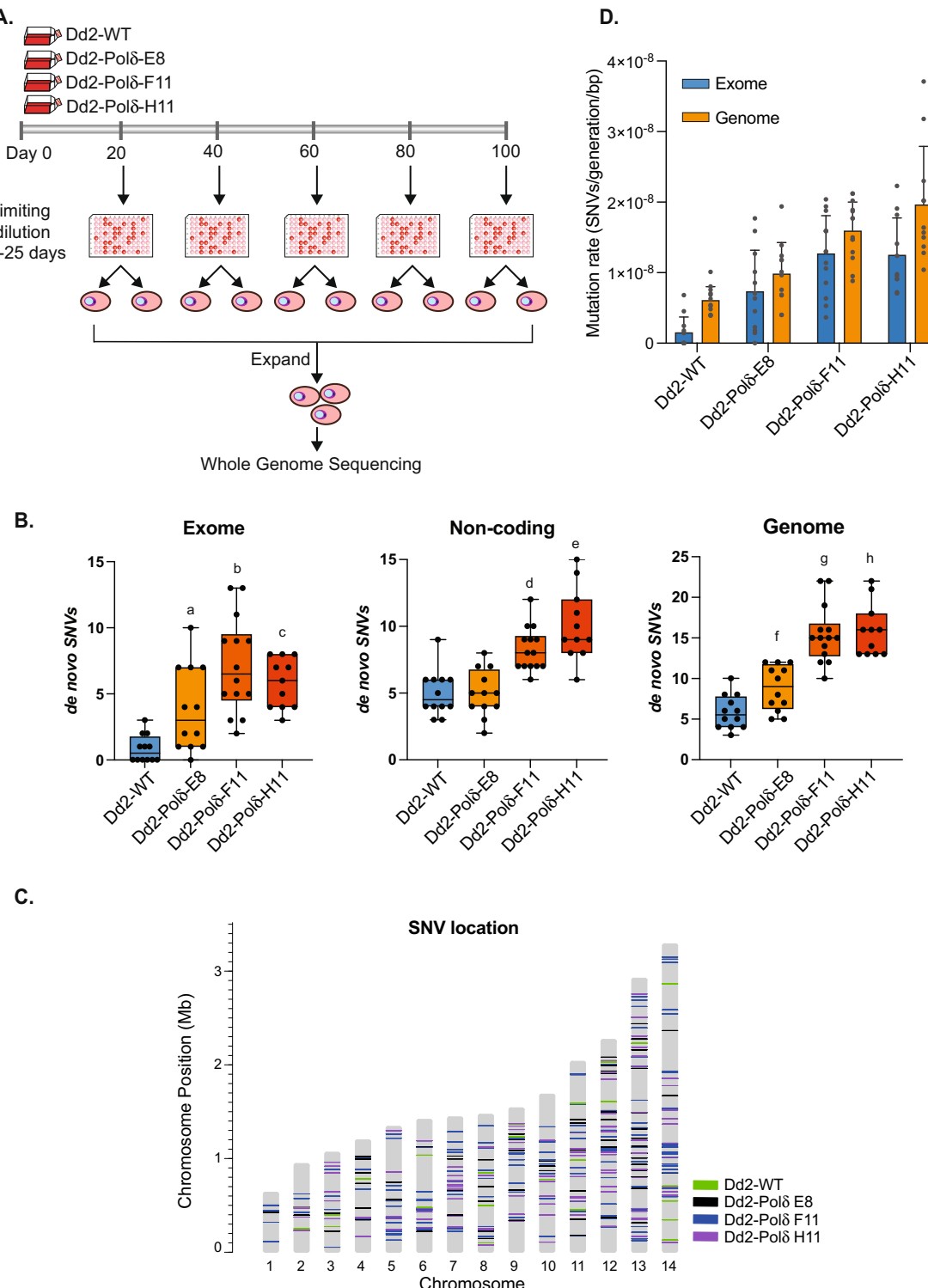

**Fig. 2 | Elevated mutation rate of the DNA polymerase δ mutant line. A** Mutation accumulation assay comparing Dd2-WT with three clones of Dd2-Polδ. All lines were cultured in parallel for 100 days (~50 generations). Parasites were sampled for clonal isolation every 20 days and subsequently harvested for genomic DNA extraction. Whole-genome sequencing was performed on samples collected on day 0, 20, 40, 60, 80 and 100. **B** The number of unique SNVs in the exome, non-coding and core genome regions of Dd2-WT and Dd2-Polδ lines were identified by subtracting from the SNVs found on day 0. Each point represents one clone, where n=the following number of independent clones (WT:12, E8:12, F11:14, H11:11).

The median line is shown and 25th/75th percentile bounded by the box, with whiskers showing min-max. A two-tailed Wilcoxon matched-pairs signed-rank test showed statistically significant differences for the Dd2-Polδ clones relative to Dd2-WT, with p values as indicated (a:0.0049; b:0.0015; c:0.002; d:0.002; e:0.002; f:0.0298; g:0.0005; h:0.001). **C** Genomic position of SNVs, colour-code by parasite line. **D** The mutation rates of Dd2-WT ($n = 12$) and Dd2-Polδ clone E8 ($n = 12$), F11 ($n = 14$) and H11 ($n = 11$), data are presented as mean +/−SD (with values and 95% confidence intervals shown in Table 1). Source data are provided as a Source Data file.

**Table 1 | Mutation rate per haploid genome or exome per generation in *P. falciparum* Dd2-WT and Dd2-Polδ parasites cultured in the absence or presence of drug pressure**

| Parasite lines | Number of clones | Mutation rate (Genome) | Mutation rate (Exome) | Fold-change (Genome) | Fold-change (Exome) |
|---|---|---|---|---|---|
| No drug pressure | | | | | |
| Dd2-WT | 12 | 6.12E-9 (4.91E-9–7.33E-9) | 1.54E-9 (1.58E-10–2.92E-9) | – | – |
| Dd2-Polδ-E8 | 12 | 9.86E-9 (7.06E-9–1.27E-8) | 7.37E-9 (3.67E-9–1.11E-8) | 1.6 | 4.8 |
| Dd2-Polδ-F11 | 14 | 1.59E-8 (1.36E-8–1.83E-8) | 1.27E-8 (9.64E-9–1.58E-8) | 2.6 | 8.2 |
| Dd2-Polδ-H11 | 11 | 1.97E-8 (1.41E-8–2.52E-8) | 1.25E-8 (9.04E-9–1.6E-8) | 3.2 | 8.1 |
| With drug pressure | | | | | |
| Dd2-WT | | | | | |
| KAE609 | 2 | 4.42E-09 (0–4.19E-8) | 4.64E-09 (0–2.99E-8) | 0.7 | 3.0 |
| Dd2-Polδ-H11 | | | | | |
| KAE609 | 3 | 3.72E-8 (2.63E-8–4.81E-8) | 4.42E-8 (1.11E-8–7.72E-8) | 6.1 | 28.7 |
| MMV665794 | 6 | 2.56E-8 (2.05E-8–3.08E-8) | 2.99E-8 (2.25E-8–3.73E-8) | 4.2 | 19.4 |
| Salinopostin A | 6 | 2.34E-8 (1.86E-8–2.82E-8) | 2.61E-8 (2.03E-8–3.19E-8) | 3.8 | 16.9 |
| KM15HA | 2 | 1.58E-8 (0–7.19E-8) | 1.93E-8 (4.88E-9–3.37E-8) | 2.6 | 12.5 |

Fold change was calculated by using untreated Dd2-WT as a comparator in both the no-drug and drug-pressured conditions. Values in brackets represent 95% confidence intervals, with negative values adjusted to zero. Underlying data are included in Supplementary Data 2.

range of starting inocula set at $2 \times 10^6$, $2 \times 10^7$, $2 \times 10^8$ and $1 \times 10^9$ parasitized cells, cultured in the presence of 2.5 nM KAE609 (~5-fold $IC_{50}$ of Dd2-WT) in three independent flasks per inoculum.

After 5 days of drug treatment, no viable parasites were detected by microscopy. Recrudescence of Dd2-WT was only observed with the highest starting inoculum of $1 \times 10^9$, with parasites observed on days 18, 21 and 30 in the three independent selection flasks. In contrast, the Dd2-Polδ line returned parasites by day 12, and with a lower starting inoculum (Fig. 3A). All three flasks with $2 \times 10^8$ and $1 \times 10^9$ parasites were positive, and one out of three flasks with $2 \times 10^7$ parasites also showed recrudescent parasites on day 12. None of the three flasks were positive for parasites with the starting inoculum of $2 \times 10^6$ (Fig. 3A).

Prior to selection, both the Dd2-WT and Dd2-Polδ parental lines had a similar $IC_{50}$ of about 0.3–0.6 nM. The KAE609-selected lines from the Dd2-WT background had $IC_{50}$ values in the range of 3–9 nM (Fig. 3B). In comparison, the drug-selected lines from the Dd2-Polδ background were appreciably more resistant with $IC_{50}$ values around 700–900 nM (Fig. 3B), three orders of magnitude higher than their parental line.

To identify the resistance determinants driving these phenotypes, the set of selected lines was examined by whole-genome sequencing as well as direct Sanger sequencing of the *pfatp4* gene. Both approaches revealed mutations in *pfatp4* (PF3D7_1211900) in these resistant lines, causing protein mutations at L350H and G199V in the Dd2-WT background from flask 1 and flask 3, respectively, and G358S in all lines from the Dd2-Polδ background (Supplementary Data 3). All three mutations are predicted to be located within or near the PfATP4 transmembrane region[33] (Fig. 3C). The mutations L350H and G358S were previously reported from an in vitro resistance experiment in Dd2 and 3D7 respectively with the dihydroisoquinolone SJ733, another compound targeting PfATP4[34]. L350H was also selected using KAE609 in a Cambodian isolate[25]. In addition to PfATP4, non-synonymous SNVs in five other genes were observed (Supplementary Data 3), illustrating that when selecting with the Dd2-Polδ line additional criteria may be required to prioritise candidates when no prior knowledge of the target is available. Of the five additional genes (PF3D7_0318500, PF3D7_0520800, PF3D7_1002200, PF3D7_1004200, PF3D7_1428400), examination of the *piggyBac* mutagenesis screen[31] identified two as non-essential, the mutation for a third gene was in a low-complexity region, and for the fourth the mutation was also observed in some clones of the mutation accumulation experiment without drug and thus is likely a background mutation.

Overall, these results confirmed that the Dd2-Polδ line can select for drug-resistant parasites in the expected molecular target[24,25,34], with lower numbers of starting parasites ($2 \times 10^7$ vs $1 \times 10^9$) and in a shorter period of selection than Dd2-WT (12 days vs 18–30 days).

### Dd2-Polδ successfully yields resistant parasites from an "irresistible" compound

We next challenged the Dd2-Polδ line with an "irresistible" compound. The "irresistible" class of compounds generally refers to compounds that fail to yield a drug-resistant parasite during in vitro selections with standard strains (typically Dd2 or 3D7). Identifying the mechanism of action of these compounds is of high interest due to their low propensity for resistance[35]. MMV665794, also known as TCMDC-124162 (2-N,3-N-bis[3-(trifluoromethyl)phenyl]quinoxaline-2,3-diamine), is a quinoxaline scaffold antimalarial identified from a phenotypic high-throughput screen[36]. Initial in vitro drug resistance selections were performed with this compound in wild-type 3D7 and Dd2 strains using different approaches, but without success (Supplementary Table 4).

We treated Dd2-Polδ and Dd2-WT with 95 nM ($1 \times IC_{50}$) of the quinoxaline compound intermittently. To maximise the chance of obtaining a resistant line, we used a high starting inoculum of $1 \times 10^9$ parasites per flask, in triplicate (Fig. 4A). After 10–12 days of pressure, no parasites could be detected by microscopy for both lines, and drug pressure was removed after day 20. Dd2-WT did not recover during the 60-day exposure period (Fig. 4A), consistent with previous unsuccessful selections (Supplementary Table 4). In contrast, all three flasks of the Dd2-Polδ line recovered on day 21. The drug concentration was then increased to $2 \times IC_{50}$, resulting in a suppression of parasites. At day 40, cultures were switched to drug-free complete medium, and on day 60, parasites were again detected in all three flasks. Clonal lines isolated from the drug-selected parasites had an increased $IC_{50}$ of about 2–2.5 fold compared with the parental line not exposed to drug pressure (Fig. 4B). The parasites from two flasks proliferated normally when re-exposed to constant drug pressure at $2 \times IC_{50}$, but parasites in the third flask did not survive.

To investigate whether higher-level resistance could be obtained, the parasite cultures of two recrudescent flasks were each split into two more flasks (each with $4.5 \times 10^8$ parasites) that were further pressured at either $3 \times IC_{50}$ or $4 \times IC_{50}$. Parasites died after 4 days in both treatments and were subsequently cultured in drug-free complete medium (Fig. 4A). However, no parasites recovered after 60 days, indicating that only low-level resistance could be obtained against MMV665794.

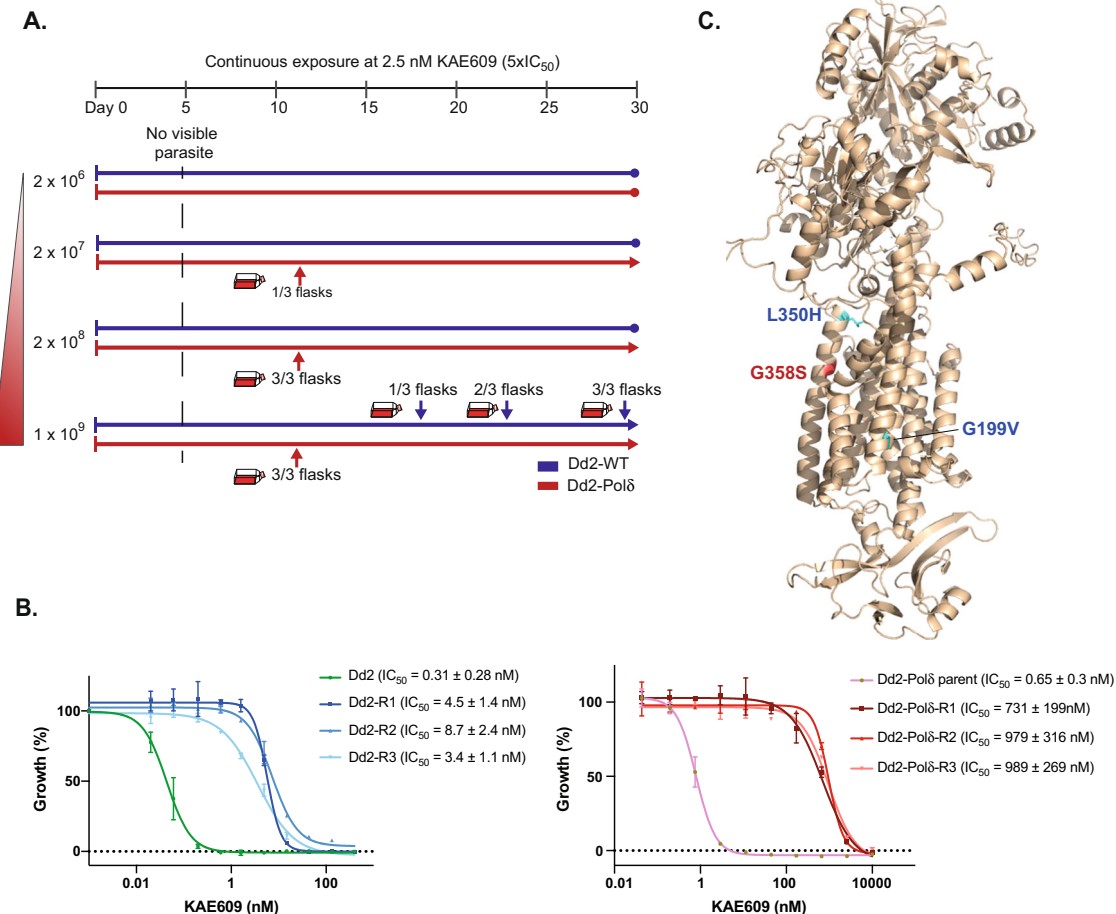

**Fig. 3 | Efficient selection of resistance to KAE609 using the DNA polymerase δ mutant parasite. A** Dd2-WT (blue line) and Dd2-Polδ (red line) were continuously cultured in the presence of 2.5 nM KAE609 ($5 \times$ IC$_{50}$). Parasite inocula ranged from $2 \times 10^6$ to $1 \times 10^9$ cells, in triplicate flasks, and parasites were detected by microscopy over the 30-day selection period. Dd2-Polδ parasites were observed on day 12 with the starting inoculum of $2 \times 10^7$, $2 \times 10^8$ and $1 \times 10^9$, whereas Dd2-WT parasites were only detected with the $10^9$ inoculum, appearing in three flasks on day 18, 21 and 30, respectively. **B** Dose-response curves of KAE609 for parental lines not exposed to drug pressure and drug-selected lines (R1-R3) for Dd2-WT (*left panel*) and Dd2-Polδ (*right panel*). Shown is a representative assay (with two technical replicates, error bars showing SD), with IC$_{50}$ values represented as mean +/−SD derived from the following biological replicates (Dd2-WT, $n = 6$; Dd2-R1, $n = 3$; Dd2-R2, $n = 3$; Dd2-R3, $n = 3$; DNA-Polδ, $n = 6$; Polδ-R1, $n = 6$; Polδ-R2, $n = 6$; Polδ-R3, $n = 5$). **C** AlphaFold model of PfATP4 showing KAE609 resistance mutations located in or near the transmembrane domains. Blue residues originated from Dd2-WT selections, red from Dd2-Polδ. Source data are provided as a Source Data file.

## Quinoxaline-resistant lines possess mutations in a gene of unknown function

To identify the causal resistance mutations in the MMV665794-selected lines (Fig. 4), we performed whole-genome sequencing on six clones isolated from two independent selections. The only mutated gene in common between all quinoxaline-selected lines was PF3D7_1359900 (PfDd2_130065800), encoding a conserved *Plasmodium* membrane protein of unknown function. The protein of 2126 amino acids encodes four predicted transmembrane domains (Fig. 5A). Each of the 6 clones contained one of two distinct SNVs, either G1612V (four clones from flask 2) or D1863Y (two clones from flask 3), equivalent to G1616V and D1864Y in Dd2, respectively (Fig. 5A and Supplementary Data 3). No new copy number variants were detected in drug-selected clones (Supplementary Data 4).

To gain some insight into the potential function of PF3D7_1359900, which only has evident orthologs within the Apicomplexa (Supplementary Fig. 5), we examined a structural model of the region containing the resistance mutations using trRosetta and AlphaFold[37,38]. This region is located towards the C-terminus of the protein, downstream of the 4 predicted transmembrane segments (Fig. 5A). Protein structure comparison using the DALI server indicated

potential structural homology with esterases/lipases, with a putative Ser-Asp-His catalytic triad located in close proximity on the protein model and highly conserved across orthologs (Fig. 5B and Supplementary Fig. 5), however further studies will be required to characterise any potential hydrolase activity.

To validate the drug-selected mutations in PF3D7_1359900, we generated CRISPR-Cas9 edited lines by introducing the Dd2 equivalent of either the G1612V or D1863Y mutation. In addition, control parasite lines were generated that were only modified with the corresponding silent mutations (G1612sil and D1863sil) and the gRNA shield mutations common to all edited lines. Both mutant lines, but not the silent controls, displayed the same modest shift in IC$_{50}$ to MMV665794 observed in the drug selected parasites (Fig. 5C).

To examine whether mutations in PF3D7_1359900 had arisen in the context of other in vitro evolution experiments, we examined the database of SNVs identified by the Malaria Drug Accelerator Consortium in 262 *P. falciparum* lines selected with 37 compounds and identified a single clone with a frameshift mutation at residue D100 of PF3D7_1359900[16]. Notably, the clone had been pressured with MMV007224, a compound with a similar quinoxaline scaffold to

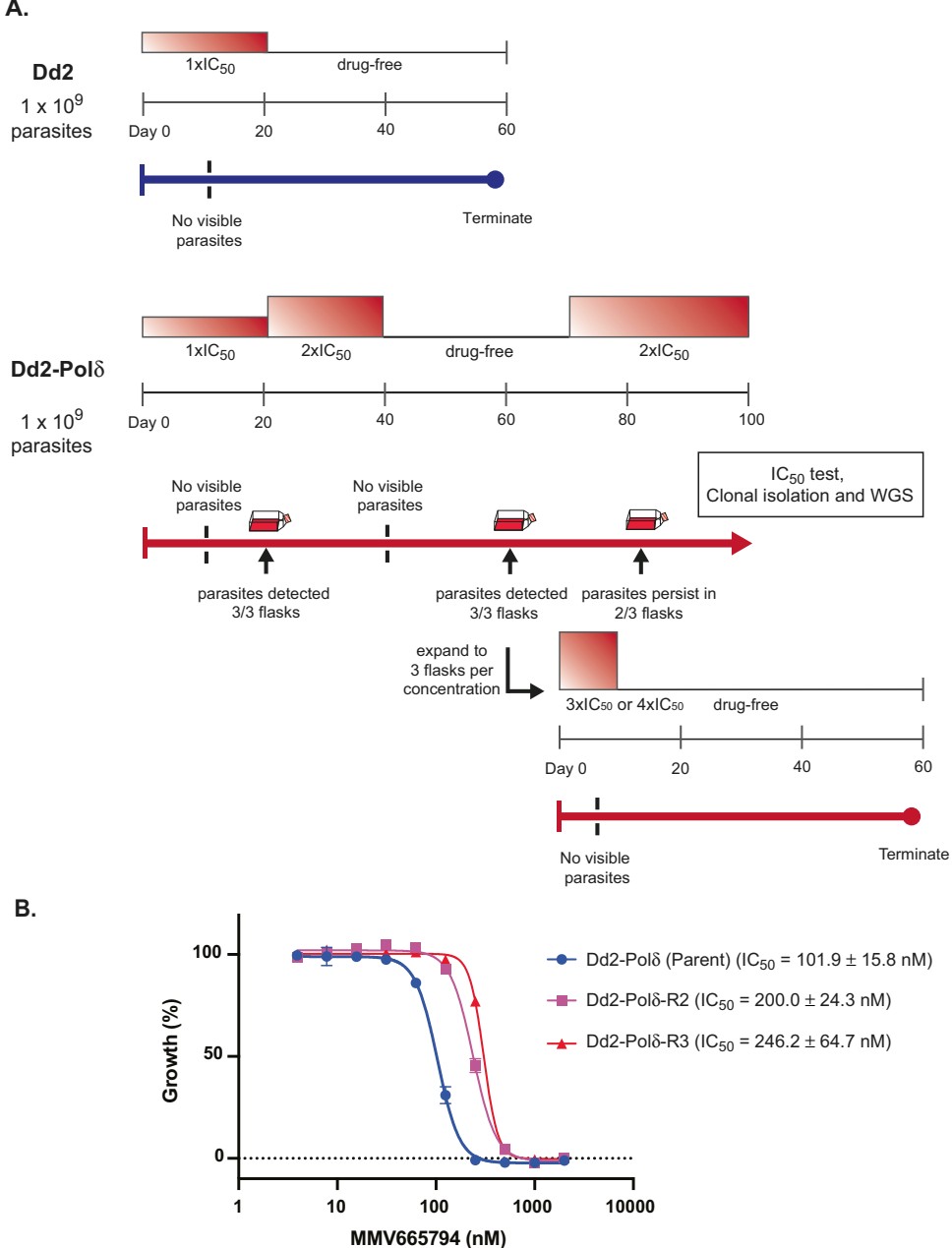

**Fig. 4 | Evolution of resistance to an "irresistible" compound. A** Selection scheme showing inability to evolve resistance to MMV665794 with Dd2-WT, but successful isolation of resistance with Dd2-Polδ. *Upper panel:* Dd2-WT exposed to $1 \times IC_{50}$ (95 nM) for 60 days. *Lower panel:* Dd2-Polδ selection pressure was ramped to $2 \times IC_{50}$ (190 nM), with recrudescence observed after 60 days in 3 independent flasks, but only 2 of 3 flasks could stably grow under drug pressure. Recrudescent parasites were further challenged with 3× and 4 × $IC_{50}$ but failed to survive. **B** Dose-response curves of MMV665794 for parental lines and the two resistant lines (C1-2) that were able to survive under $2 \times IC_{50}$ pressure. Shown is a representative assay (two technical replicates, error bars showing SD), with $IC_{50}$ values represented as mean values +/−SD derived from the following biological replicates (DNA-Polδ, $n = 7$; Polδ-R1, $n = 5$; Polδ-R2, $n = 5$). Source data are provided as a Source Data file.

MMV665794 (Fig. 5D). The presence of a frameshift mutation near the start of the protein (Supplementary Fig. 6A), plus mutagenesis in the *piggyBac* whole-genome screen[31] indicates this gene is non-essential during the asexual blood stage.

We tested whether the CRISPR-edited parasites bearing the MMV665794-resistance mutations could confer cross-resistance to its analogue, MMV007224. Both the G1612V and D1863Y mutant lines showed a similar low-level resistance to MMV007224 as observed with MMV665794 (Fig. 5D). Likewise, the MMV007224-selected line bearing the frameshift mutation in PF3D7_1359900 displayed equivalent low-grade resistance to MMV665794, but not unrelated compounds KAE609 and chloroquine (Supplementary Fig. 6B).

These findings suggest that the protein encoded by PF3D7_1359900, which we have designated as quinoxaline-resistance protein 1 (PfQRP1), may confer general resistance to quinoxaline-like compounds. To explore whether mutations in PfQRP1 mediate resistance more broadly against compounds with a quinoxaline-like scaffold, we synthesised six analogues of MMV665794 (Supplementary Material) and commercially sourced an additional five analogues, evaluating this panel against the QRP1-D1863Y CRISPR-edited parasite. All but one quinoxaline analogue showed low-level resistance in the QRP1-mutant line significantly different from the controls, in contrast to BQR695 and KDU691[39], antimalarials targeting phosphatidylinositol 4-kinase expected to have an unrelated mode of action to the

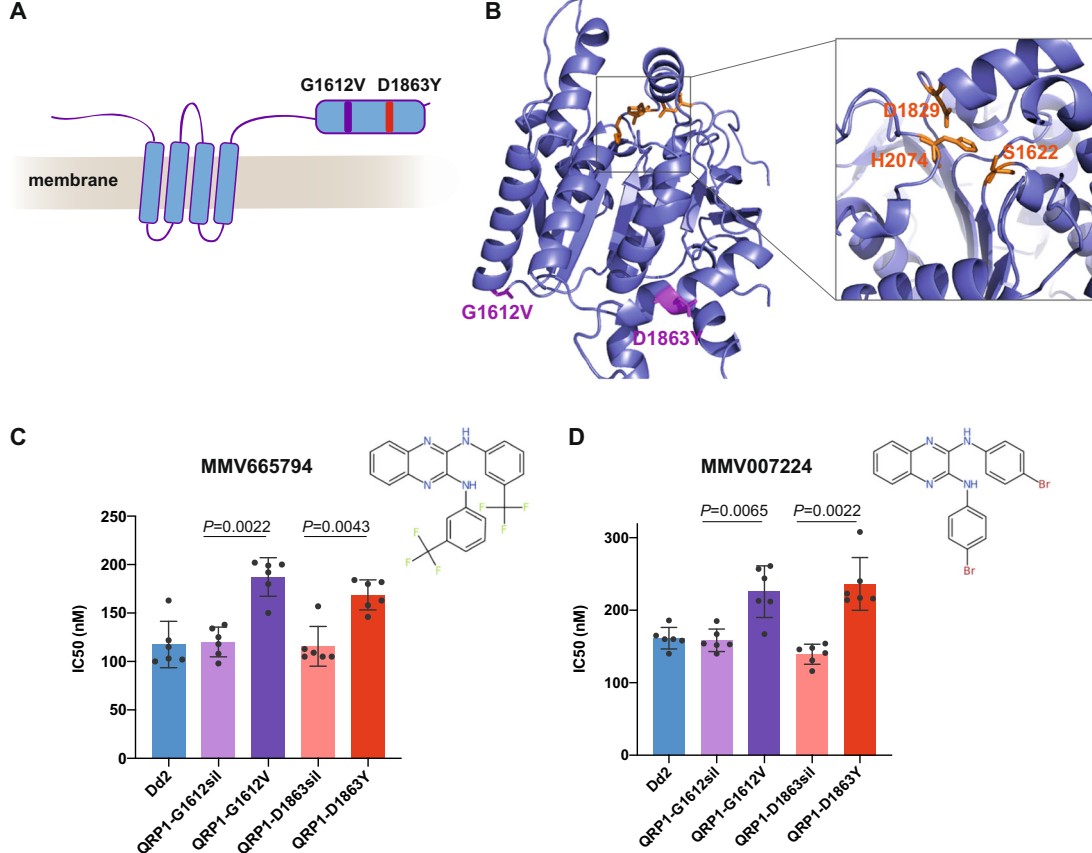

**Fig. 5 | QRP1 confers resistance to quinoxaline compounds. A** PfQRP1 (PF3D7_1359900) encodes a 250 kDa protein with four predicted transmembrane domains. The two mutations G1612V and D1863Y found in two independent selections with MMV665794 are located in a C-terminal domain that shares conservation with α/β hydrolases. **B** Model of the C-terminal 656 residues (1471–2126) of PfQRP1 showing the putative catalytic triad of Ser-Asp-His (yellow), and the G1612V and D1863Y resistance mutations (purple). **C, D** IC$_{50}$ values of CRISPR-Cas9 edited QRP1. Dd2 lines encoding the equivalent G1612V and D1863Y mutations showed a significantly reduced susceptibility against MMV665794 and were cross-resistant to MMV007224, a structurally related molecule sharing the quinoxaline scaffold, in comparison with Dd2-WT and silent edited controls. Each dot represents a biological replicate, with $n = 6$ independent replicates with mean ± SD shown, and statistical significance relative to silent-edited controls determined by two-sided Mann–Whitney $U$ test. Source data are provided as a Source Data file.

quinoxaline compounds (Supplementary Fig. 7). Our data suggest that PfQRP1 is a previously uncharacterised protein that confers low-grade resistance to quinoxaline-based compounds.

## Dd2-Polδ under drug pressure has an elevated mutation rate

The augmented ability of the Dd2-Polδ line to generate resistance to both KAE609 and MMV665794 was consistent with an increase in genetic diversity available for selection. We determined the mutation rate of Dd2-Polδ under drug pressure, comparing parasites selected with KAE609, MMV665794, and two additional resistance selections with the unrelated irresistible compounds Salinopostin A and KM15HA[40,41]. De novo SNVs of the drug-selected lines were determined in comparison with the parental lines used in the corresponding batch of selection experiments (Fig. 6A and Supplementary Fig. 8). The change of mutation rate in these lines was compared with non-pressured Dd2-WT, reflecting the combined factors of a defective proof-reading DNA polymerase δ and drug pressure. Dd2-Polδ under drug pressure displayed an increased mutation rate in coding regions of 13–28 fold, and ~3–6 fold in the genome relative to non-drug pressured wild-type Dd2 (Fig. 6B and Table 1). When compared with non-drug treated Dd2-Polδ, these changes translate to an increase of ~1.5–3.5 fold in coding regions and essentially unchanged (~0.8–1.9 fold) across the genome. The Ts:Tv ratio of Dd2-Polδ under drug pressure was varied and did not show a discernible trend (Supplementary Fig. 9).

The mutation rate of Dd2-WT under drug pressure also increased ~3 fold in coding regions relative to non-drug pressured Dd2-WT. However, this was relatively unchanged across the whole genome (Table 1), consistent with previous reports[18].

Collectively, our data indicate that the Dd2-Polδ line has an increased mutation rate that provides enhanced potential of selecting drug-resistant parasites, even with previously irresistible compounds, while being sufficiently low to maintain genome integrity and parasite robustness.

## Discussion

We propose that the Dd2-Polδ mutator *P. falciparum* parasite is a powerful new tool to identify targets and resistance mechanisms of antimalarials. The defective proof-reading resulting from the engineered modification to DNA polymerase δ results in an increased rate of spontaneous mutation. By expanding the genetic sequence space in cultured parasites, we reveal an enhanced capability to yield drug-tolerant lines under in vitro evolution of resistance regimes. We observed that for selections with a drug with a known mode-of-action, KAE609[24], we obtained resistant parasites with 10–100 fold lower inoculum and in a shorter selection window using the Dd2-Polδ line. In the case of an irresistible compound, the quinoxaline MMV665794, the Dd2-Polδ line yielded modestly resistant parasites where previously selections had failed. One potential consideration arising from the elevated mutation rate is the presence of more unrelated genetic

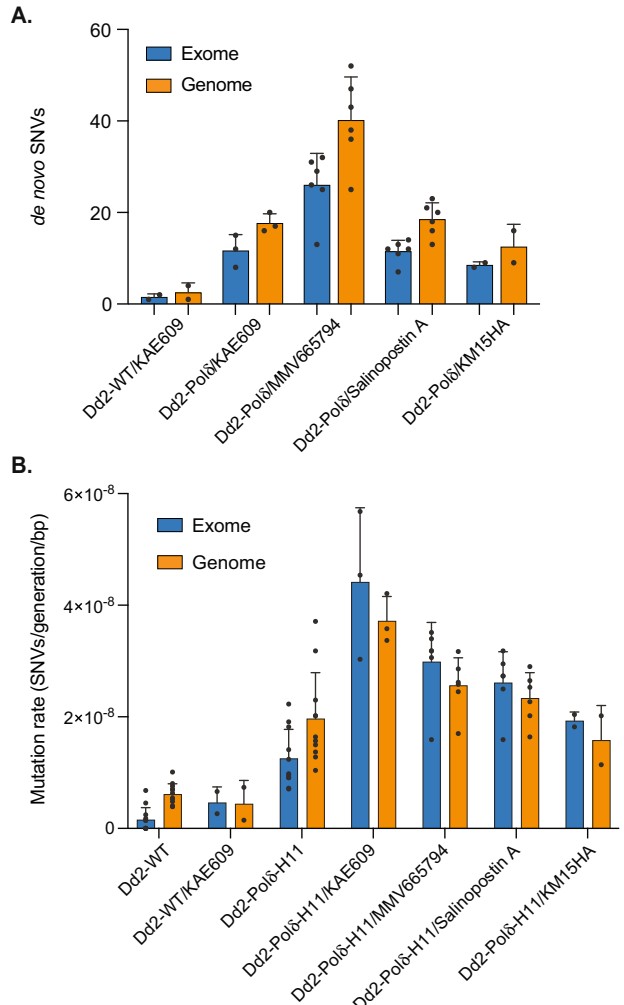

**Fig. 6 | Increased number of SNVs and mutation rate of Dd2-Polδ under drug pressure. A** The number of SNVs in Dd2-Polδ selected with different antimalarial compounds. These selections, except for KAE609, failed to yield resistant parasites with Dd2-WT. Note the higher number of SNVs in KAE609-selections with Dd2-Polδ compared with Dd2-WT (see Supplementary Table 6). Each dot represents a sequenced clone, with the number of independent clones in brackets: Dd2-WT + KAE609 (2); Dd2-Polδ+KAE609 (3); Dd2-Polδ+MMV665794 (6); Dd2-Polδ+SalA (6); Dd2-Polδ+KM15HA (2). The blue and orange bars represent the exome and core genome, respectively, with means ± SD shown. **B** The mutation rates of Dd2-WT and Dd2-Polδ under drug pressure. Each dot represents a sequenced clone, with the number of independent clones under drug as noted for (**A**) above, and as follows: Dd2-WT-drug (12); Dd2-Polδ-drug (11). Data are presented as mean +/−SD (see Table 1 for values and 95% confidence intervals). Data from non-drug treated Dd2-WT and Dd2-Polδ clone H11 (see Fig. 2D) are included for comparison. Source data are provided as a Source Data file.

mutations occurring during drug resistance selection. Sequencing at least two clones from more than one independent selection, coupled with genome editing validation, will therefore be important for pinpointing causal mutations.

Using a mutation accumulation assay combined with whole-genome sequencing allowed us to determine a mutation rate based on whole-genome data, not dependent on representative reporter loci[42]. We followed wild-type and Dd2-Polδ parasites over 100 days to derive a mutation rate. For Dd2-Polδ parasites, this rate was approximately 3-fold higher than Dd2-WT across the genome, and up to 8-fold when comparing changes in the exome, with the caveat that the mutation rate for the wild-type exome may be underrepresented as 7 of 13 parental Dd2 clones acquired no SNVs over the time frame of the

experiment. Thus Dd2-Polδ requires a smaller number of parasites for a mutation to occur in its haploid genome than Dd2-WT (5.08E7 vs 1.63E8 parasites; Supplementary Data 2).

In the presence of antimalarial compounds, the mutation rate across the exome was 1.5–3.5 fold higher in the Dd2-Polδ line relative to the non-treated line. Although one explanation may be the potential mutagenic stress of the treatments themselves, the mutation rate across the whole genome was only modestly increased (0.8–1.9 fold) when compared with non-drug treated lines, consistent with a previous study that found a less than 3-fold mutation rate increase under atovaquone selection[18]. This seeming increase within the exome may thus reflect the positive selection of functional mutations that impact the ability to survive drug pressure or to maintain fitness by supporting primary resistance mutations. Alternatively, clones that accumulate higher numbers of bystander mutations may be selected against in the absence of drug pressure, but would outcompete clones with a lower mutation burden if they possess a fortuitous resistance SNV.

The mild mutator phenotype of Dd2-Polδ may be advantageous in two respects, by not creating too many mutations under selection to allow identification of the likely causal mutations, and by maintaining fitness despite the potential generation of detrimental mutations. In comparison with Dd2-WT, the Dd2-Polδ line showed only a minor loss of fitness, and we did not observe reversion of the engineered D308A/E310A mutations in DNA polymerase after long-term culture. In contrast, the equivalent DNA polymerase δ mutant in *P. berghei* showed a significant reduction in fitness, and the presence of an antimutator mutation in DNA polymerase δ was observed[22,23]. The higher mutation rate of the *P. berghei* DNA polymerase δ mutant, approximately 90-fold over wild-type, and potentially the more stringent growth conditions in vivo may explain the greater impact on parasite fitness in the rodent malaria parasite. No mutations of these two residues on DNA polymerase δ were found in clinical isolates existing in the Pf6K database[43].

Antimutator mutations in DNA polymerases act to increase fidelity and can themselves have inherent fitness costs (Herr et al., 2011). We did not observe known antimutator mutations in DNA polymerase δ in any of the sequenced *P. falciparum* Dd2-Polδ clones, perhaps a reflection of the limited selective pressure imposed by the moderate elevation in mutation rate. Nonetheless, all three Dd2-Polδ clones possessed SNVs in or near genes that play roles in DNA replication and DNA repair, although whether these mutations confer functional effects is unknown. The non-coding SNV close to proliferating cell nuclear antigen 2 (PCNA2) in clone H11 (Supplementary Data 1) may potentially modulate gene expression of *pcna2*, one of two PCNA proteins in *P. falciparum*, to facilitate high processivity of DNA polymerase δ[44–46]. In addition, Dd2-Polδ clone E8, which displayed a lower mutation rate than the other two clones (F11 and H11), has a non-synonymous SNV (G435E) in the putative DNA polymerase theta (PF3D7_1331100) Gly435, equivalent to Gly226 in human DNA polymerase θ that lies in the region of the DEAD/DEAH box helicase. DNA polymerase θ possesses a low fidelity DNA polymerase and helicase activity, and plays a role in DNA repair such as double-strand break repair through canonical non-homologous end joining, microhomology-mediated end joining and homologous recombination. Polymerase θ has an impact on genome stability and repairing breaks formed by G4 quadruplex structures[46,47].

The ability of the Dd2-Polδ line to elicit resistant parasites was evaluated using two compounds, KAE609 (cipagarmin) and MMV665794. KAE609, currently in Phase II clinical trials, targets the P-type ATPase PfATP4 that is responsible for transport of Na$^+$ across the parasite plasma membrane (Rottmann et al., 2010; Spillman et al., 2013). All three mutations obtained from our selections were in the predicted transmembrane region of PfATP4, consistent with most previously observed mutations (Rottmann 2010; Jimenez-Diaz et al., 2014; Viadya et al., 2015; Lee and Fidock, 2016). Notably, selections with the Dd2-Polδ line yielded a G358S mutant that was recently

observed in the majority of treatment failures during a Phase II trial of KAE609 (Schmitt et al., 2021), indicating that in vitro evolution with this line can yield outcomes with clinical relevance. This high-level resistance mutation was also observed previously in selections with a dihydroisoquinolone compound (+)-SJ733 (Jimenez-Diaz et al., 2014), as well as in parallel KAE609 selections with our Dd2-Polδ line by another group[48]. Qui et al. generated a CRISPR-edited line with the G358S SNV, demonstrating that this mutation is sufficient to drive a large shift in IC$_{50}$ and showed that it protects the Na$^{+}$-ATPase activity of PfATP4 from inhibition by KAE609, but at the cost of lowering the affinity of the protein to Na$^{+}$[48]. Despite this functional effect on PfATP4, the mutant parasites do not have a strong fitness cost[48]. This may partly explain why we only observed the G358S mutant in selections with the Dd2-Polδ line as it may outcompete slower growing or less resistant mutants in the bulk culture.

In addition to obtaining more facile resistance with lower parasite numbers, the main potential of the Dd2-Polδ line is in accessing new sequence space for previously irresistible compounds. We challenged the Dd2-Polδ line with MMV665794, an irresistible compound with a quinoxaline chemical scaffold and a flutamide-like functional group[16,49]. Our selections with Dd2-Polδ yielded a modestly resistant parasite after approximately 60 days, whereas selections with wild-type 3D7 or Dd2 lines failed (Supplementary Table 4)[16].

Whole-genome sequencing of MMV665794-resistant clones revealed mutations in PF3D7_1359900 in all six clones selected from two independent selection flasks. The gene is predicted to be non-essential based on a single piggyBac insertion site approximately 0.8 kb into the 7 kb gene[31] as well as our observation of a frameshift mutant near the start of the gene. CRISPR-Cas9 editing of the G1612V and D1863Y mutations confirmed their role in the resistance phenotype. Furthermore, these parasites displayed cross-resistance to several structurally related quinoxaline-like compounds. The protein encoded by PF3D7_1359900, which we have termed quinoxaline resistance protein 1 (PfQRP1), is predicted to contain 4 transmembrane domains towards the N-terminus, and a domain towards the C-terminus within which the resistance mutations are located that shares conservation with conventional hydrolases. The non-essential nature of PfQRP1 suggests this is not the target of the quinoxaline compounds but a resistance mechanism. Although we do not know whether the G1612V and D1863Y mutations confer loss of function, the presence of the frameshift mutation in the MMV007224-selected line is suggestive. Whether QRP1 acts directly on quinoxaline compounds, in a manner akin to the PfPARE esterase that converts a prodrug to an active form[50] or has an indirect effect is not known. Nonetheless, the level of resistance elicited is modest with only a two-fold loss of potency. Thus, the difficulty in obtaining resistance to MMV665794, together with the limited shift in potency, suggest that compounds of this chemical class may be promising antimalarial candidates for further exploration. More generally, as more previously "irresistible" compounds are explored using the Dd2-Polδ line, it will be interesting to determine if this approach more frequently yields mutations in resistance genes rather than direct targets. The nature of "irresistible" compounds may mean that mutation of the direct target results in a non-viable parasite, or the compound may hit multiple targets or targets of host origin. These possibilities may lead to resistance mutations as the most likely outcome.

Evolution of resistance in vitro coupled with whole-genome analysis has proven to be a highly successful technique for understanding the mechanism of action of novel compounds as well as identifying markers for drug resistance in the field[13,51]. One limitation of this approach has been the relative difficulty of eliciting resistance to some chemical classes. By increasing the genetic complexity of in vitro cultures, the Dd2-Polδ parasite line developed here has the potential to reduce the parasite inoculum, accelerate the selection time, and enable exploration of previously irresistible compounds.

## Methods

### Genome editing using CRISPR-Cas9

*P. falciparum* Dd2 strain was employed for all genetic manipulations using CRISPR-Cas9. To generate the "mutator" line, the conserved catalytic amino acid residues D308 and E310 of DNA polymerase δ catalytic subunit (PF3D7_1017000) were mutated to alanine. Two different single guide RNAs (sgRNA) and a donor repair template (length of 699 bp, with 363 bp/289 bp flanking homology) harbouring the double D308A/E310A mutations with additional shield mutations to prevent sgRNA-Cas9 complex binding were cloned sequentially into a single plasmid that contains SpCas9 and the hdhfr selectable marker (see Fig. 1A)[52]. To introduce the putative resistance mutations G1612V and D1863Y into PF3D7_1359900, two sgRNAs and a donor repair template for each mutation were constructed as above. For G1612V a donor template of 746 bp (443 bp/235 bp flanking homology) was generated, and for D1863Y a donor template of 671 bp (252 bp/307 bp flanking homology) was generated. Guide RNAs were synthesised as oligo primers (IDT). Donor repair templates and control donor templates encoding only silent mutations at the targeted sites were synthesised by GeneArt (Thermo Fisher Scientific). The 5′ and 3′ of donor DNA templates were flanked by an additional 20–21 bp sequence with homology to the destination pDC2-Cas9-gRNA plasmid for insertion at the AatII and EcoRI sites using NEBuilder HiFi DNA Assembly[52]. The plasmid constructs were verified by Sanger sequencing. The sgRNAs and Sanger sequencing primers are shown in Supplementary Table 5.

### Parasite cultivation and transfection

Parasites were cultured in RPMI 1640 (Gibco) complete medium consisting of 0.5% Albumax II (Gibco), 25 mM HEPES (culture grade), 1x GlutaMAX (Gibco), 25 µg/mL gentamicin (Gibco), and supplied with O$^{+}$ human red blood cells (RBCs) obtained from anonymous healthy donors from National Health Services Blood and Transplant (NHSBT) or Red Cross (Madrid, Spain). The use of RBCs was performed in accordance with relevant guidelines and regulations, with approval from the NHS Cambridgeshire Research Ethics Committee and the Wellcome Sanger Institute Human Materials and Data Management Committee for the experiments performed in the UK. RBCs were sourced ethically and their research use was in accord with the terms of the informed consents under an IRB/EC approved protocol for experiments done in Spain. Parasites were routinely maintained at 0.5–3% parasitemia with 3% hematocrit and were cultured under malaria gas (1% O$_2$, 3% CO$_2$ and 96% N$_2$). A synchronous ring stage was obtained by 5% sorbitol treatment in the cycle prior to electroporation. In the next cycle, the ring stage (0–16 h) at 10% parasitemia was harvested for electroporation. The pDC2-Cas9-gRNA-donor plasmid was transfected into the *P. falciparum* Dd2 line using a Gene Pulser Xcell (BioRad). Plasmids containing either sgRNA1 or sgRNA2 (50 µg) or a mix of both plasmids (100 µg) were mixed with 150 µl of packed parasitised-infected red blood cells and complete cytomix (120 mM KCl, 0.2 mM CaCl2, 2 mM EGTA, 10 mM MgCl2, 25 mM HEPES, 5 mM K2HPO4, 5 mM KH2PO4; pH 7.6) to make a total volume of 420 µl[52]. The transfectants were selected in complete medium containing 5 nM WR99210 (Jacobus Pharmaceuticals) for 8 days. The culture was subsequently maintained in drug-free complete medium until parasites reappeared. Limiting dilution was performed to isolate clonal gene-edited parasites (Fig. 1B). Transfectants from bulk and clonal cultures were genotyped by allele-specific PCR and Sanger sequencing. Primers are shown in Supplementary Table 5.

### Mutation accumulation assay

The mutation accumulation assay was performed with Dd2-WT and three Dd2-Polδ clones. Mixed-stage parasites at 1–5% parasitemia in 10 ml were cultured continuously for 100 days. Parasites were taken out of the continuous cultures on days 0, 20, 40, 60, 80, and 100 for clonal isolation by limiting dilution in 96-well plates. One to three

parasite clones from each time point were propagated for genomic DNA extraction by DNeasy Blood & Tissue Kits (Qiagen) for whole-genome sequencing on a Hiseq X (Illumina).

## Competitive fitness assay

The assay was performed by mixing the test and control parasites at a 1:1 ratio with 1% parasitemia each. Dd2-GFP, a Dd2 line expressing green fluorescent protein from the ER hsp70 promoter[32], was used as the reference parasite that competed against either Dd2-WT or Dd2-Polδ in a 6-well plate. The hematocrit of the query and the competitor cultures was determined using the Cellometer Auto 1000 (Nexcelom Bioscience). Parasitemia was determined by staining parasites with MitoTracker Deep Red FM (Invitrogen) and counting using a CytoFlex S flow cytometer (Beckman Coulter) running software CytExpert v2.4 and analysed further using FlowJo v10, with counts and parasite stage confirmed by microscopic examination following Giemsa staining (VWR Chemicals). Uninfected RBCs were used as a signal background for gating on the flow cytometer. The competitive fitness was determined by measuring the total parasitemia by MitoTracker Deep Red staining, and the proportion of GFP-positive control parasites on the flow cytometer every two days for 20 days (about 10 generations). Samples were prepared in a 96-well round-bottom plate (Costar) by taking 4 μL of culture into 200 μL phosphate buffer saline (PBS) (Gibco) containing 100 nM of Mitotracker Deep Red FM. The plate was incubated at 37 °C for 15 min and subjected to analysis on the flow cytometer. The gates were set up for the FITC (gain 5 or 10) and APC (gain 3 or 5) channels for GFP and Mitotracker Deep Red FM signals, respectively. A representative gating strategy is shown in Supplementary Fig. 10. Two independent biological experiments with three technical replicates were performed.

## In vitro drug resistance selections using Dd2-Polδ

Two compounds were used for in vitro evolution of resistance, KAE609 (cipargamin) and MMV665794, an antimalarial compound identified in the Tres Cantos Antimalarial Set and included in the Medicines for Malaria Venture Malaria Box[36,49]. To determine the minimum inoculum for resistance (MIR) for KAE609, three independent flasks containing ring stage cultures of Dd2-WT and Dd2-Polδ clone H11 were tested at a range of inocula ranging from $2 \times 10^6$, $2 \times 10^7$, $2 \times 10^8$, and $1 \times 10^9$ parasites. Each flask was continuously cultured in complete medium containing 2.5 nM ( ~ $5 \times IC_{50}$) of KAE609. This concentration was able to kill parasites to a level undetectable by microscopy of Giemsa-stained thin smears (minimum of 30 fields of 100 RBCs). Parasite death and recrudescence after drug treatment was monitored by Giemsa staining of thin smears, with microscopic examination every one to two days. Selections with MMV665794 were performed with Dd2-WT and Dd2-Polδ with intermittent drug exposure in three independent flasks (illustrated in Fig. 4A). Parasites at an initial inoculum of $1 \times 10^9$ were continuously exposed to 95 nM of MMV665794 ($1 \times IC_{50}$) for 20 days. Then, Dd2-WT was maintained in drug-free complete medium until day 60. Dd2-Polδ parasites that reappeared after selection were subsequently exposed to a two-fold increment of MMV665794 at 190 nM until day 40. Drug pressure was removed until parasites were detected, and the concentration was thereafter ramped up to $3 \times IC_{50}$ and $4 \times IC_{50}$ with an inocula of $4.5 \times 10^8$ parasites. For all selected lines, parasite clones were isolated by limiting dilution and propagated for 18–25 days. Parasites before drug pressure and surviving parasites after drug pressure were harvested for genomic DNA extraction and whole-genome sequencing.

## Drug susceptibility assay

Drug susceptibility assays were performed in 96-well plates using synchronized ring-stage parasites prepared by using 5% sorbitol. The ring stage parasites in the next cycle were diluted to 1% parasitemia with 2% hematocrit (final concentration in the assay plate) to perform

the half-maximal growth inhibitory concentration assay ($IC_{50}$). The concentration range was prepared by two-fold serial dilution of compound in complete medium. Parasites untreated or treated with 5 μM artesunate and RBCs only (2% hematocrit) were included in the assay plate as controls. Parasite growth was determined after 72-h drug incubation by using 2x lysis buffer containing 2 × SYBR Green I (Molecular Probes)[53] and measurement on a FluorStar Omega v5.11. $IC_{50}$ analysis was performed using GraphPad Prism v8/v9 and statistical significance was determined by two-sided Mann–Whitney $U$ tests. All assays were performed in three to six independent biological experiments with two technical replicates each.

## Whole genome sequencing

Parasite samples were lysed in 0.1% saponin, washed with 1×PBS, and genomic DNA (gDNA) was extracted using the QIAamp DNA Blood Midi Kit (Qiagen). The concentration of gDNA was quantified using the Qubit dsDNA BR assay kit and measured with a Qubit 2.0 fluorometer (Thermo Fisher Scientific) prior to sequencing. The samples were sheared to around 450-bp fragments and the library constructed using the NEB-Next UltraII DNA library kit (NEB), followed by qPCR for sample pooling and normalisation for the Illumina sequencing platform. Paired-end sequencing (2 × 150 bp) and PCR-free whole genome sequencing was performed on a HiSeq X (Illumina)[54]. Samples selected for resistance to Salinopostin A and KMHA15 were sequenced on an Illumina MiSeq or NextSeq 550 sequencing platform, respectively, to obtain 300 or 150 bp paired-end reads at an average of 30× depth of coverage.

## Single nucleotide variant and copy number variant calling

The genome sequences of Dd2-Polδ clones were analysed by following the GATK4 best practice workflow[55]. Paired-end sequencing reads from each parasite clone were aligned to *P. falciparum* 3D7 (PlasmoDB-44_Pfalciparum3D7) and Dd2 reference sequence (PlasmoDB-44_PfalciparumDd2) using bwa mem (bwa/0.7.17 = pl5.22.0_2). PCR duplicates were removed by GATK MarkDuplicates (picard/2.22.2--0) (Supplementary Table 6). Variant calling was performed by GATK Haplotype-Caller (gatk/4.1.4.1). The SNVs had to pass the filtering criteria (ReadPosRankSum ≥ −8.0, MQRankSum ≥ −12.5, QD ≥ 20.0, SOR ≥ 3.0, FS ≤ 60.0, MQ ≥ 40.0, GQ ≥ 50.0, DP ≥ 5.0). Variants that had heterozygous calls or were located outside of the core genome were excluded (the Dd2 core genome coordinates are shown in Supplementary Table 1). Genetic variant annotation and functional effect prediction were determined by using snpEff (v4.3t)[56]. Transcription start sites were mapped according to the recent refined data set[57]

The number of de novo SNVs occurring during the mutation accumulation assay was identified by using the genome of Dd2-WT and Dd2-Polδ collected on Day 0 for subtraction in each parasite line. For the drug pressure condition, the de novo SNVs were identified by using the genome of the parental line that was not exposed to drug pressure for subtraction. The significant change of SNV numbers in Dd2-WT and Dd2-Polδ in the condition without drug was determined by two-sided Wilcoxon matched-pairs signed-rank tests (GraphPad Prism v8/v9).

CNVs were detected by the GATK 4 workflow[58] adapted for *P. falciparum*[59]. Briefly, read counts were collected across the gene-encoding regions of the *P. falciparum* core genome[60] and denoised log$_2$ copy ratios were calculated against a panel of normals constructed from non-drug-selected Dd2 samples. CNVs were retained if at least 4 sequential genes showed a denoised log$_2$ copy ratio greater than or equal to 0.5 (copy number increase) or less than or equal to −0.5 (copy number decrease).

## Mutation rate determination

The mutation rate (μ) of each parasite line was determined by the mean number of de novo single nucleotide variants (S) from all clones (C) that occurred during continuous parasite culture and that differed from the parasite line on Day 0. The duration of erythrocytic life cycles

(L) and Genome size (G) were calculated as shown below[20,21]. A single asexual blood stage cycle for Dd2 was calculated at 44.1 h[20]. The sizes of the Dd2 core genome and coding region were set as 20,789,542 bp and 11,553,554 bp, respectively (Supplementary Data 2). Shapiro-wilk normality test was used to examine SNV datasets for normal distribution. One sample t-test was used to examine mean samples and 95% confidence intervals (Supplementary Data 2). All tests were run by R programming (v4.1.3) and data collated in Microsoft Excel (v16).

$$\mu = \frac{\frac{\Sigma(\frac{S}{T})}{\Sigma C}}{G}$$

## Protein structure modelling

The protein structures of PfATP4 (PF3D7_1211900) and PfQRP1 (PF3D7_1359900) were modelled by AlphaFold[37] and comparisons performed using the DALI server[61]. Structures were displayed using PYMOL molecular graphics system.

## Reporting summary

Further information on research design is available in the Nature Portfolio Reporting Summary linked to this article.

## Data availability

The data underlying this article are available within the supplementary material and source data files. All associated sequence data are available at the NCBI Sequence Read Archive under accession code ERP110649 (BioProject: PRJEB28444). Library names DN581642P:A7, D7, E7 and DN573783H:A5-12, B5-B12, C5-C12, D5-D8, D10-D12, E5-E12, F5-F12, G6-G7, G9-G12, H4-H12. Reference genomes for Dd2 and 3D7 and gene annotation data are available on PlasmoDB (Dd2 at https://plasmodb.org/common/downloads/release-44/PfalciparumDd2/; 3D7 at https://plasmodb.org/common/downloads/release-44/Pfalciparum3D7/). Source data are provided with this paper.

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

## Acknowledgements
We would like to thank current and former members of the Lee lab for constructive feedback. We are grateful to Liz Huckle and the staff in Sanger Scientific Operations for their support with sequencing. We thank Kotanan N. for artwork. We would like to acknowledge funding from the Bill and Melinda Gates Foundation to M.C.S.L., D.A.F., G.S.K., and E.A.W. (OPP1054480), and to D.A.F. (INV-033538), funding from the Wellcome Institutional Translational Partnership award (ITPA) and Mahidol University to T.K. and T.C., and funding from Wellcome [206194/Z/17/Z] to M.C.S.L. and the Wellcome Sanger Institute.

## Author contributions
K.K. and M.C.S.L. conceived the study. K.K. generated the Dd2-Polδ parasite lines and performed the mutation accumulation experiments. K.K. and M.R. performed parasite transfections. K.K., K.A.S., S.M.C., V.F., and M.G.G. performed the in vitro drug selection experiments. Drug sensitivity and cross-resistance assays were performed by K.K., J.O., M.R., K.A.S., and M.C.S.L. H.P. and A.C.U. contributed to the generation of whole-genome sequencing data. G.P.J., B.C.M., B.L., E.C. and J.D. contributed to compound synthesis. Whole-genome sequencing data were analysed by K.K., T.K., T.Y., M.R.L., J.H., R.P., and S.M. K.K., F.J.G., E.A.W., D.A.F., T.C. and M.C.S.L. conceptualized and planned the experiments, and supervised the study. All authors contributed to writing the paper.

## Competing interests
The authors declare no competing interests.
