## [Peer Review File · Nature Communications]

Reviewer comments, initial review

Reviewer #1 (Remarks to the Author):

The manuscript by Kümpornsin et al describes the generation of a transgenic Dd2 parasite line that has 2 mutations in the DNA polymerase δ that impairs its proof reading activity. As a consequence, this leads to an increase in nucleotide sequence variation and higher mutation rates, which the authors have then capitalised on for in vitro drug resistance selection experiments. This has enabled them to obtain parasites resistant to "irresistible" compounds, which could not previously been obtained using wildtype parasites.

The experiments were robustly conducted, the conclusions sound and the manuscript well written. The experimental schematics also provided a nice overview of the experimental design which made it easy to follow. Whilst there are other strategies for target identification such as CETSA, etc, in vitro drug evolution of drug resistance and whole genome analysis of resulting clones has been especially helpful for defining modes of action. That the mutations are indeed responsible for resistance can be validated by inserting the mutation into parasites by CRISPR/Cas9. Hence the generation of this line will be of benefit to the malaria community.

There were only a few questions and minor comments I had with respect to the manuscript.

1. With the pol mutant, if there are a lot more background mutations, would the authors recommend to sequence more clones (compared to wildtype) to pinpoint the causal mutation? Is there a minimum they would recommend based on their results?
2. Fig S2. With the exception of F11, the other two clones did not have an increased de novo SNV over time. Is this not surprising?
3. Is it also surprising that in the absence of drug pressure that there was a higher fold change for mutations in the exome, when compared to the exome? Why would this be?
4. Given that the compound MMV665794 was an irresistible compound, yet resistance selection on the Dd2-Pol line yielded mutations in a non-essential gene PfQRP1, this does indeed suggest this gene is not the target, rather a resistance mechanism as the authors suggest in the discussion. It will be interesting to test other irresistible compounds and see whether this is a general phenomenon or whether increasing the frequency of mutations will enable target ID. (I'm not suggesting to do more experiments here, it's just a comment). Nevertheless, identifying potential resistance mechanisms will be useful for analysing drug resistance in the field.
5. Supp Fig 3 – It is a bit hard to discern from the figure whether there was any preference for the type of variant. Perhaps this could be mentioned in the text. Were there any particular mutation hotspots? It appears that there were differences between the clones in where the SNVs were so I guess not.

Minor comments:

Abstract: It is not apparent what KAE609 is. It should be mentioned that this is a compound and what the target is.

Table S9 is referred to first in the text and hence should be Table S1. Table S1 referred to in the text is actually Table S8.

Table S5 is not referred to in the text.

Supp Fig 2b should be in a separate figure.

Methods: It would be good to provide more details on the length of homology on either side of mutations in the donor template.

Reference 49 needs amending.

Reviewer #2 (Remarks to the Author):

Comments to required items from the editor;

1. What are the noteworthy results?

The authors have constructed a mutator *P. falciparum* parasite (hereafter mutator Pf) by editing DNA polymerase delta. The mutator Pf showed a 5–8 fold elevation of mutation rate. Importantly, despite the knockdown of important parasite function, the mutator Pf displayed only small growth disadvantage compared to parent parasite line. Two features of mutator Pf (moderate mutation rate and lack of significant fitness loss) would be favorable to perform laboratory evolutionary study to make (select) a parasite with particular phenotype such as drug resistance. They indeed have shown that KAE609 resistant Pf parasites were able to be constructed by mutator Pf under the pressure of it in a shorter time and with smaller initial parasite volume than by Dd2 wild-type. They have also exhibited that selections were successful with an “irresistible” compound, MMV665794 that failed to yield resistance. However, only moderate resistant parasites were able to be selected. This might not be a main point of this study, but may further support the “irresistibility” of MMV665794.

2. Will the work be of significance to the field and related fields?

By using mutator Pf, we are able to easily evaluate how easy resistant parasite to new drug will emerge. This makes it easier to choose “proper compounds” as candidate of antimalarial drug in advance.

3. How does it compare to the established literature? If the work is not original, please provide relevant references.

In 1991, it was reported that 3' to 5' exonuclease activity located in the DNA polymerase delta subunit was required for accurate replication in *Saccharomyces cerevisiae*. Transgenic yeast with the two amino acids that are critical for proofreading in DNA polymerase showed higher mutation rate (Simon M, EMBO, 1991). In 2014, generation of mutator Pb parasite (Honma H, DNA Res, 2014) and usefulness of mutator Pb for rapid construction of Pb with decreased piperazine sensitivity (Ikeda M, Front Microbiol, 2021).

4. Does the work support the conclusions and claims, or is additional evidence needed?

This work supports the conclusion, and no additional evidence is not required.

5. Are there any flaws in the data analysis, interpretation and conclusions? Do these prohibit publication or require revision?

No flaw was observed in this study.

6. Is the methodology sound? Does the work meet the expected standards in your field?

The methodologies used in this study are reasonable and enough to make conclusions.

7. Is there enough detail provided in the methods for the work to be reproduced?

Method is fully described to reproduce the work by other researchers.

Suggestions and questions to the authors;

As mentioned above, this is a wonderful work potentially to speed up the development of novel antimalarial drugs. Followings are comments/suggestions.

1. In the current version, mutation rates were determined after non-synonymous and synonymous mutation were mixed. It would be interesting to determine the mutation rate of non-synonymous and synonymous separately.

2. It is very interesting that mutation rates substantially increased under the drug selection. Actually, mutator Pf under drug pressure showed 1.5–3.5 fold in coding regions compared with non-drug treated mutator Pf. The authors described the possible reason for it as “positive selection of functional mutations that impact the ability to survive drug pressure or to maintain fitness by supporting primary resistance mutations”. It is plausible, but I am afraid that this can explain a part of mutations. If all these mutations are related to drug resistance directly (increase in resistant level) or indirectly (compensation of potential fitness lost), mechanisms of drug resistance may be very complicated. With the current data set, it may not be easy, but if possible, could you comment on this?

L50: "Furthermore, artemisinin resistance is a public health threat to people living in endemic regions worldwide, as exemplified by recent reports of the emergence of Kelch13 mutations in Rwandan and Ugandan isolates that cause reduced artemisinin susceptibility"

Two reports are cited in this sentence from Uganda (Balikagala, NEJM 2021) and Rwanda (Uwimana, Nat Med 2020). However, the report from Uganda has shown the evidence of emergence of clinical artemisinin resistance as observed in SEA. So, "the emergence of Kelch13 mutations in Rwandan and Ugandan isolates that cause reduced artemisinin susceptibility" is misleading.

L199: Supplementary Table 4 appears to be error. Could you describe all SNPs that were shared three KAE609 clones? From Supplementary Table 6, it looks there are six gens with non-synonymous SNPs.

L451: Just confirmation. Was parasite culture performed without human serum (only Albmax II) in all experiments (transfection, competitive fitness assay, drug selection and determination of IC50 by in vitro drug susceptibility assay)?

L511: How may RBC did count to determine parasite negative by microscopic examination with thin smears?

Reviewer #3 (Remarks to the Author):

This manuscript describes a generation of the hyper mutator *Plasmodium falciparum*, of which DNA polymerase δ is mutated, and successful identification of drug resistance gene using this mutator. The similar experiment in the rodent malaria parasite, *P. berghei*, had been already reported, and thus the concept is not novel. However, applying this identification system for drug resistance using mutators to the most deadly human malaria parasite, *P. falciparum*, is important achievement. The author could identify the PfATPase4 mutation, G358S which is considered to confer the KAE609, using this system. The selected parasite clone with this mutation possesses the strong drug resistance, which is 1,000 fold greater, compared to wild-type parasite. Interestingly, this mutation was also found in the failure of phase II trial of KAE609, showing the usefulness of this system using the mutator. In addition, they demonstrate that the drug resistance parasites to "irresistible" drug, such as MMV665794, could be produced efficiently using the mutator. Genome-wide SNV analysis of the obtained drug resistance parasite allowed to identify the mutation of PF3D7_1359900, PfQRP1, which is responsible for drug resistance.

I agree with the authors that this is a useful system for identifying the drug resistance gene. However, I think that the further characterization of identified mutations, which are G358S in PfATPase4, G1616V and D1864Y in PfQRP1, are needed. In addition, several minor revision are also needed.

Major comment

1: The mutation, G358S, in PfATPase4 is reported to be highly related with the failure of phase II trial of KAE609 (Clin Infect Dis. 2022 May 30;74(10):1831-1839. doi:10.1093/cid/ciab716.). Interestingly, the similar mutation was found in the selected parasite using the mutator, and the result strongly suggested that it possibly conferred strong resistance, with which IC50 was 400 – 600 nM, to the parasite. I think this finding relationship between mutation and treatment failure is clinically important and will be an experimental evidence for proving the usefulness of this system using mutator. For further clarify the usefulness of the system, the transgenic parasite with this mutation shall be generated and whether it can confer observed strong resistance shall be confirmed.

2: The authors predicted based on the structural comparison that PfQRP1 has hydrolytic activity, because it has the putative catalytic triad, which is consist of Ser-Asp-His. Furthermore, they assessed the drug resistance of the transgenic parasite with mutations G1612 and D1863Y to

MMV665794, MMV007224, MMV665852, and MMV011438 and suggested based on obtained results that PfQRP1 is a non-essential putative hydrolase that confers resistance to quinoxaline-based compounds. I agree those mutation are responsible for drug resistance. However, the amino acid conservation and predicted structure are not sufficient to describe that it is a putative hydrolase; it may be possible to say that it shares sequence conservation with conventional hydrolases, such as serine proteases. In addition, I would like to ask whether they have an idea about involvement of hydrolytic activity of PfQRP1 in drug resistance. The author examined drug resistance to MMV011438, which is hydrolase-susceptible compound, and showed that the transgenic parasite did not show any resistance to it. I think based on this result that it is hard to say that hydrolytic activity is responsible for drug resistance. Furthermore, if author state that PfQRP1 is the drug resistance gene to "quinoxaline-based compound", they shall examine many other drugs in addition to MMV665794 and MMV007224. Experimental evidences to only two compounds is not sufficient for this statement.

3: The author found the frameshift at residue D100 of PfQRP1 in the public database. This mutation was found in the parasite clone which is exposed to drug pressure with MMV007224. This suggested that parasite may acquire the drug resistance by losing the function of PfQRP1. Since the PfQRP1 is predicted to be a dispensable gene for asexual development, it is possible to disrupt it by CRISPR/Cas9. This gene disruption will be useful for understanding the drug resistance mechanism using PfQRP1. This experiment is one of approach for validating causal mutation found by the system using mutators, and thus should be carried out.

Minor comments

1: The similar experimental concept had already been proved in the rodent malaria parasites. Therefore, the appropriate description should be included in the introduction section.

2: The result of analysis about base pair substitutions for transition (Ts) and transversion (Tv) was described at line 151 – 153. What is the aim for this analysis?

3: From line 155, the author describe about the mutation rate of Dd2-Pol δ . Please briefly explain how you calculate the mutation rate. Although I know that it is described in the method section, it will assist readers for understanding the efficiency of mutating in Dd2-Pol δ .

4: Dd2-Pol δ clone H11 does not have any obvious mutation in CDS of molecules, which are involved in DNA repair, and exhibit best mutation rate among mutator clones. Do you considered that the mutations found in clone E8 and F11 may interfere with DNA repair? If so, please explain the reason and discuss about it.

5: In line 186, the drug resistance parasites to KAE609 were emerged by day12. I think one of the reason is that the parasite with G358S acquired the strong resistance with IC50 value 400 – 600 nM. Although the parasite exhibiting weak and moderate drug resistance would emerge with same frequency as the parasite with G358S, they might become minor population in the culture due to slower multiplication in the presence of drug. As a result, it appear that only parasites with G358S may appear to have generated from mutator lines. Please discuss about this possibility.

6: In line 349 - 351, the authors described the possibility about the greater mutation rate in exons than that in intergenic regions in the presence of drug. This may be examined using the transgenic parasite. For example, if the transgenic parasite with G358S shows worse growth than the parasite selected from mutator in the presence of drug, selected parasite acquired the additional mutation which assists survival. On the other hand, if the transgenic parasite with G358S shows worse growth than wild type in the absence of drug, this mutation impose the fitness cost to the parasites. This experiment may be useful for understanding the strategy of the parasite acquiring the mutation in exons.

Response to Reviewer Comments

Reviewer #1 (Remarks to the Author):

The manuscript by Kümpornsinn et al describes the generation of a transgenic Dd2 parasite line that has 2 mutations in the DNA polymerase δ that impairs its proof reading activity. As a consequence, this leads to an increase in nucleotide sequence variation and higher mutation rates, which the authors have then capitalised on for in vitro drug resistance selection experiments. This has enabled them to obtain parasites resistant to “irresistible” compounds, which could not previously been obtained using wildtype parasites.

The experiments were robustly conducted, the conclusions sound and the manuscript well written. The experimental schematics also provided a nice overview of the experimental design which made it easy to follow. Whilst there are other strategies for target identification such as CETSA, etc, in vitro drug evolution of drug resistance and whole genome analysis of resulting clones has been especially helpful for defining modes of action. That the mutations are indeed responsible for resistance can be validated by inserting the mutation into parasites by CRISPR/Cas9. Hence the generation of this line will be of benefit to the malaria community.

Reply: Thank you for the thoughtful evaluation and comments, which we have endeavoured to answer in detail below.

There were only a few questions and minor comments I had with respect to the manuscript.

1. With the pol mutant, if there are a lot more background mutations, would the authors recommend to sequence more clones (compared to wildtype) to pinpoint the causal mutation? Is there a minimum they would recommend based on their results?

Reply: To account for the presence of more background mutations in the Dd2-Pol δ mutant, we have now included a suggestion in the Discussion (page 12, line 363-365) to sequence a minimum of two clones per selection, from multiple independent selections, and to use CRISPR validation to aid with identifying the causal mutation.

2. Fig S2. With the exception of F11, the other two clones did not have an increased de novo SNV over time. Is this not surprising?

Reply: We agree that there is not a simple linear increase rate in detected SNVs over the time course. We think that this may reflect the relatively long time it would take to generate large numbers of mutations, even with the Dd2-Pol δ mutant line, and that extended sampling and culture over much longer time frames (e.g. >1 year) would provide a clearer trajectory. Nonetheless to help the reader visualize the numbers obtained over the 100 day assay and to also benchmark with expected numbers based on mutation rate, we have plotted the total SNVs observed and expected in the genome and exome in a new Supplementary Figure 2B.

3. Is it also surprising that in the absence of drug pressure that there was a higher fold change for mutations in the exome, when compared to the exome (*reviewer meant genome?*)? Why would this be?

Reply: This is a good point - we agree that a large difference between the genome and exome rates might not be expected, particularly in the absence of selective pressure. However, this may partly reflect the fact that we observed very few, and in several samples 0 SNVs in the wild type exome, due to the low mutation rate of the wild type parasite. Thus the calculated mutation rate for the wild type exome may appear artificially low because of this. The rate for the whole genome, which being larger accumulated more SNVs even in the wild type Dd2 line, may be more representative. We have now noted this caveat in the Discussion (page 12, line 370-373).

4. Given that the compound MMV665794 was an irresistible compound, yet resistance selection on the Dd2-Pol line yielded mutations in a non-essential gene PfQRP1, this does indeed suggest this gene is not the target, rather a resistance mechanism as the authors suggest in the discussion. It will be interesting to test other irresistible compounds and see whether this is a general phenomenon or whether increasing the frequency of mutations will enable target ID. (I'm not suggesting to do more experiments here, it's just a comment). Nevertheless, identifying potential resistance mechanisms will be useful for analysing drug resistance in the field.

Reply: This will be an interesting point to review once more examples of resistance selections of irresistible compounds have been performed with the Dd2-Pol δ line. Nonetheless, it does appear that selections with MMV665794 and the Salinopostin A and KM15HA compounds all yielded likely resistance mechanisms rather

than direct targets, with SaIA selections identifying a PRELI domain protein that may function in multidrug resistance (Yoo et al., 2020, PMID: 31978322) and KM15HA yielding mutations in the multidrug transporters *mdr2* and *mdr1* (Knabb et al., 2021). The nature of “irresistible” compounds may mean that either mutation of the direct target results in a non-viable parasite, or the compound may hit multiple targets. Both possibilities may lead to resistance mutations as the most likely outcome. We have included this point in the Discussion (page 15, line 463-468).

5. Supp Fig 3 – It is a bit hard to discern from the figure whether there was any preference for the type of variant. Perhaps this could be mentioned in the text. Were there any particular mutation hotspots? It appears that there were differences between the clones in where the SNVs were so I guess not.

Reply: There is an increase in particular of non-synonymous variants. Although we do not have an explanation for this observation, we have now noted this in the text (page 6, line 161-163) and shown in Supplementary Figure 3B. For hotspots, for each line (Dd2 wild type and 3 Dd2-Pol δ I clones) we first examined the frequency of mutations per chromosome (relative to chromosome length) and did not find that these differed from the expected proportions. We also examined whether the same or a nearby (<20bp) mutation occurred in different lines, and identified 12 examples that are now shown in a new Supplementary Table 4. These are all in highly repetitive AT-rich intergenic regions, for which mapping confidence is lower. We also identified four genes that had multiple distinct non-synonymous mutations (although these were not <20bp in proximity), and have added this to Supplementary Figure 3C. We have briefly summarised these points on page 6, line 163-166.

Minor comments:

Abstract: It is not apparent what KAE609 is. It should be mentioned that this is a compound and what the target is.

Reply: We have noted that this is a spiroindolone PfATP4 inhibitor.

Table S9 is referred to first in the text and hence should be Table S1. Table S1 referred to in the text is actually Table S8.

Reply: We have revised the arrangement of our Tables accordingly.

Table S5 is not referred to in the text.

Reply: We have removed this table (but included the data in the source material) as this is shown in Figure 6A.

Supp Fig 2b should be in a separate figure.

Reply: We have move this to a separate figure, Supp Fig. 8.

Methods: It would be good to provide more details on the length of homology on either side of mutations in the donor template.

Reply: We have included these details now in the Methods section (page 17, section “Genome editing using CRISPR-Cas9”).

Reference 49 needs amending.

Reply: We have updated this BioRxiv manuscript with the published paper and updated the text.

Reviewer #2 (Remarks to the Author):

Comments to required items from the editor;

1. What are the noteworthy results?

The authors have constructed a mutator *P. falciparum* parasite (hereafter mutator Pf) by editing DNA polymerase delta. The mutator Pf showed a 5–8 fold elevation of mutation rate. Importantly, despite the knockdown of important parasite function, the mutator Pf displayed only small growth disadvantage compared to parent parasite line. Two features of mutator Pf (moderate mutation rate and lack of significant fitness loss) would be favorable to perform laboratory evolutionary study to make (select) a parasite with particular phenotype such as drug resistance. They indeed have shown that KAE609 resistant Pf parasites were able to be constructed by

mutator Pf under the pressure of it in a shorter time and with smaller initial parasite volume than by Dd2 wild-type.

They have also exhibited that selections were successful with an “irresistible” compound, MMV665794 that failed to yield resistance. However, only moderate resistant parasites were able to be selected. This might not be a main point of this study, but may further support the “irresistibility” of MMV665794.

2. Will the work be of significance to the field and related fields?

By using mutator Pf, we are able to easily evaluate how easy resistant parasite to new drug will emerge. This makes it easier to choose “proper compounds” as candidate of antimalarial drug in advance.

3. How does it compare to the established literature? If the work is not original, please provide relevant references.

In 1991, it was reported that 3' to 5' exonuclease activity located in the DNA polymerase delta subunit was required for accurate replication in *Saccharomyces cerevisiae*. Transgenic yeast with the two amino acids that are critical for proofreading in DNA polymerase showed higher mutation rate (Simon M, EMBO, 1991). In 2014, generation of mutator Pb parasite (Honma H, DNA Res, 2014) and usefulness of mutator Pb for rapid construction of Pb with decreased piperazine sensitivity (Ikeda M, Front Microbiol, 2021).

4. Does the work support the conclusions and claims, or is additional evidence needed?

This work supports the conclusion, and no additional evidence is not required.

5. Are there any flaws in the data analysis, interpretation and conclusions? Do these prohibit publication or require revision?

No flaw was observed in this study.

6. Is the methodology sound? Does the work meet the expected standards in your field?

The methodologies used in this study are reasonable and enough to make conclusions.

7. Is there enough detail provided in the methods for the work to be reproduced?

Method is fully described to reproduce the work by other researchers.

Suggestions and questions to the authors;

As mentioned above, this is a wonderful work potentially to speed up the development of novel antimalarial drugs. Followings are comments/suggestions.

Reply: Thank you for the thoughtful consideration and comments.

1. In the current version, mutation rates were determined after non-synonymous and synonymous mutation were mixed. It would be interesting to determine the mutation rate of non-synonymous and synonymous separately.

Reply: As the SNV numbers for the wild type parasites are low, with no SNVs observed in the exome in some clones, we felt it may be difficult to obtain robust values for the separate non-synonymous and synonymous mutation rates. However, to give a sense of these differences with the Dd2-Pol δ line, we have included a new figure (Supp. Fig 3b) showing total SNVs obtained in the mutation accumulation experiment, across four categories – intergenic, synonymous, non-synonymous and nonsense. We note the relative increase in non-synonymous mutations in particular, although we do not have an explanation for this observation.

2. It is very interesting that mutation rates substantially increased under the drug selection. Actually, mutator Pf under drug pressure showed 1.5–3.5 fold in coding regions compared with non-drug treated mutator Pf. The authors described the possible reason for it as “positive selection of functional mutations that impact the ability to survive drug pressure or to maintain fitness by supporting primary resistance mutations”. It is plausible, but I am afraid that this can explain a part of mutations. If all these mutations are related to drug resistance directly (increase in resistant level) or indirectly (compensation of potential fitness lost), mechanisms of drug resistance may be very complicated. With the current data set, it may not be easy, but if possible, could you comment on this?

Reply: This is an interesting point, and it is true that it can be hard to disentangle what are “functional” mutations that might contribute to resistance or fitness or other ways to support a primary resistance mutation, and what may be unrelated bystanders. One possibility that might explain the higher number of SNVs under drug treatment (aside from potential mutagenic/stress effects of the treatment itself) is that parasites bearing a higher burden of mutations may be selected against due to fitness cost in the absence of drug, but under drug pressure the lucky parasite that has a protective resistance mutation will outcompete all other parasites even if it disadvantaged with a higher number of bystander mutations as well. We have added this speculative explanation in the Discussion

(page 13, line 377-387). We have also reworded the Discussion to note the fold change comparing treated and untreated Dd2-Pol δ clone H11 (which is 1.5 – 3.5 fold as you note), rather than the wild type parasite for which there would be a higher fold shift.

L50: “Furthermore, artemisinin resistance is a public health threat to people living in endemic regions worldwide, as exemplified by recent reports of the emergence of Kelch13 mutations in Rwandan and Ugandan isolates that cause reduced artemisinin susceptibility”

Two reports are cited in this sentence from Uganda (Balikagala, NEJM 2021) and Rwanda (Uwimana, Nat Med 2020). However, the report from Uganda has shown the evidence of emergence of clinical artemisinin resistance as observed in SEA. So, “the emergence of Kelch13 mutations in Rwandan and Ugandan isolates that cause reduced artemisinin susceptibility” is misleading.

Reply: We have replaced the Uwimana et al. 2020 reference with an updated study (Uwimana et al., 2021; PMID: 33864801) with clinical relevance.

L199: Supplementary Table 4 appears to be error. Could you describe all SNPs that were shared three KAE609 clones? From Supplementary Table 6, it looks there are six genes with non-synonymous SNPs.

Reply: We have corrected the reference to Supplementary Table 6. We have also noted now that five other genes with non-synonymous SNPs were observed. In the case of KAE609, we tested this as a proof-of-principle, and expected to identify mutations in the target PfATP4. This example provides a useful illustrative point that other considerations may be required to prioritize SNVs if no prior knowledge is available. We have added this point to the Results (page 8, line 224-232). Of the five additional genes, examination of the *piggyBac* mutagenesis screen (Zhang et al., Science 2018) identifies two as non-essential, for the third the mutation is in a low-complexity region, and for the fourth the mutation is also observed in some clones of the mutation accumulation experiment without drug and thus is likely a background mutation. Thus, it would be possible to deprioritize 4 of the 6 candidate genes for initial validation.

L451: Just confirmation. Was parasite culture performed without human serum (only Albumax II) in all experiments (transfection, competitive fitness assay, drug selection and determination of IC50 by in vitro drug susceptibility assay)?

Reply: That is correct, only Albumax II was used for all culture experiments.

L511: How may RBC did count to determine parasite negative by microscopic examination with thin smears?

Reply: A minimum of 30 fields of 100 RBCs were scanned. We have added this detail to the Methods (page 19, line 559-561).

Reviewer #3 (Remarks to the Author):

This manuscript describes a generation of the hyper mutator *Plasmodium falciparum*, of which DNA polymerase δ is mutated, and successful identification of drug resistance gene using this mutator. The similar experiment in the rodent malaria parasite, *P. berghei*, had been already reported, and thus the concept is not novel. However, applying this identification system for drug resistance using mutators to the most deadly human malaria parasite, *P. falciparum*, is important achievement. The author could identify the PfATPase4 mutation, G358S which is considered to confer the KAE609, using this system. The selected parasite clone with this mutation possesses the strong drug resistance, which is 1,000 fold greater, compared to wild-type parasite. Interestingly, this mutation was also found in the failure of phase II trial of KAE609, showing the usefulness of this system using the mutator. In addition, they demonstrates that the drug resistance parasites to “irresistible” drug, such as MMV665794, could be produced efficiently using the mutator. Genome-wide SNV analysis of the obtained drug resistance parasite allowed to identify the mutation of PF3D7_1359900, PfQRP1, which is responsible for drug resistance.

I agree with the authors that this is a useful system for identifying the drug resistance gene. However, I think that the further characterization of identified mutations, which are G358S in PfATPase4, G1616V and D1864Y in PfQRP1, are needed. In addition, several minor revision are also needed.

Major comment

1: The mutation, G358S, in PfATPase4 is reported to be highly related with the failure of phase II trial of KAE609 (Clin Infect Dis. 2022 May 30;74(10):1831-1839. doi:10.1093/cid/ciab716.). Interestingly, the similar mutation was found in the selected parasite using the mutator, and the result strongly suggested that it possibly conferred strong resistance, with which IC50 was 400 – 600 nM, to the parasite. I think this finding relationship between

mutation and treatment failure is clinically important and will be an experimental evidence for proving the usefulness of this system using mutator. For further clarify the usefulness of the system, the transgenic parasite with this mutation shall be generated and whether it can confer observed strong resistance shall be confirmed.

Reply: We agree this is a particularly interesting mutant to have obtained, given the potential clinical relevance. The recently accepted study by Qiu *et al* (PMID: 36180431), on which we collaborated, examined the mechanistic impact of G358S on PfATP4 function and reports the generation of CRISPR-edited G358S mutants. These were shown to confer a large (>750-fold) shift in IC₅₀, indicating that this single point mutation can confer high-level resistance. We have cited this study in our revised Discussion (page 14, line 432-438).

2: The authors predicted based on the structural comparison that PfQRP1 has hydrolytic activity, because it has the putative catalytic triad, which is consist of Ser-Asp-His. Furthermore, they assessed the drug resistance of the transgenic parasite with mutations G1612 and D1863Y to MMV665794, MMV007224, MMV665852, and MMV011438 and suggested based on obtained results that PfQRP1 is a non-essential putative hydrolase that confers resistance to quinoxaline-based compounds. I agree those mutation are responsible for drug resistance. However, the amino acid conservation and predicted structure are not sufficient to describe that it is a putative hydrolase; it may be possible to say that it shares sequence conservation with conventional hydrolases, such as serine proteases.

Reply: We have modified the relevant places in the Results, Discussion and Figure legend (page 10, line 287-288; page 15, lines 453-457; page 29 figure legend) to soften the wording and indicate that the domain shares conservation with hydrolases as suggested, and that further studies are required to demonstrate any hydrolase activity.

In addition, I would like to ask whether they have an idea about involvement of hydrolytic activity of PfQRP1 in drug resistance. The author examined drug resistance to MMV011438, which is hydrolase-susceptible compound, and showed that the transgenic parasite did not show any resistance to it. I think based on this result that it is hard to say that hydrolytic activity is responsible for drug resistance.

Reply: We do not as yet have clear evidence demonstrating hydrolytic activity is involved in drug resistance. We have attempted to generate mutants in the catalytic triad, but to date have been unable to obtain these CRISPR-edited lines. Thus, we cannot demonstrate whether mutations in these conserved putative catalytic residues would also confer resistance. We agree that the finding with MMV011438 is not in itself compelling evidence and have removed this data, replacing this supplemental figure with a much more extensive examination of related quinoxaline analogs (see below).

Furthermore, if author state that PfQRP1 is the drug resistance gene to “quinoxaline-based compound”, they shall examine many other drugs in addition to MMV665794 and MMV007224. Experimental evidences to only two compounds is not sufficient for this statement.

Reply: We agree it would be more compelling to test a wider number of quinoxaline-like compounds, and have appreciably expanded the panel of compounds tested. Working with the Medicines for Malaria Venture and TCG Lifesciences, we synthesised six analogues of MMV665794 (described in the Supplementary Material) and have commercially sourced an additional five analogues. All but one of these showed a significant low-level increase in IC₅₀ in the QRP1-D1836Y CRISPR edited line (see Supplementary Figure 7).

3: The author found the frameshift at residue D100 of PfQRP1 in the public database. This mutation was found in the parasite clone which is exposed to drug pressure with MMV007224. This suggested that parasite may acquire the drug resistance by losing the function of PfQRP1. Since the PfQRP1 is predicted to be a dispensable gene for asexual development, it is possible to disrupt it by CRISPR/Cas9. This gene disruption will be useful for understanding the drug resistance mechanism using PfQRP1. This experiment is one of approach for validating causal mutation found by the system using mutators, and thus should be carried out.

Reply: We have attempted to disrupt QRP1 using two different CRISPR strategies, engineering a stop codon in the C-terminal domain, and insertion of a Blasticidin S Deaminase selectable marker within the gene. However, despite multiple repeated attempts we have not been able to obtain a disrupted line. Nonetheless, to address this question as best we can, we were able to source the original frameshift mutant line, which was selected in the 3D7 background. We confirmed by Sanger sequencing the presence of the frameshift mutation (Supplementary Figure 6A) and evaluated the response of this line to the two quinoxalines MMV665794 and MMV007224, as well as two unrelated compounds, KAE609 and chloroquine (Supplementary Figure 6B). This line with the QRP1 frameshift showed a similar shift in IC₅₀ to the quinoxaline compounds as the CRISPR-edited point mutants and no shift with the unrelated control compounds,

Minor comments

1: The similar experimental concept had already been proved in the rodent malaria parasites. Therefore, the appropriate description should be included in the introduction section.

Reply: Yes, we were inspired by this approach and have noted this now in the Introduction (page 4, line 102-103), in addition to the existing references to this work in the Results and Discussion.

2: The result of analysis about base pair substitutions for transition (Ts) and transversion (Tv) was described at line 151 – 153. What is the aim for this analysis?

Reply: We examined this to determine whether there was a bias in the type of mutation observed.

3: From line 155, the author describe about the mutation rate of Dd2-Pol δ . Please briefly explain how you calculate the mutation rate. Although I know that it is described in the method section, it will assist readers for understanding the efficiency of mutating in Dd2-Pol δ .

Reply: We have added a brief description in the Results and also referred readers to the Methods for a more detailed description (page 6, line 171-173).

4: Dd2-Pol δ clone H11 does not have any obvious mutation in CDS of molecules, which are involved in DNA repair, and exhibit best mutation rate among mutator clones. Do you considered that the mutations found in clone E8 and F11 may interfere with DNA repair? If so, please explain the reason and discuss about it.

Reply: Due to the difference in mutation rate between the three clones, we examined other DNA repair genes to see if there were differences. We do not know if these might have functional effects to partially restore stringency, but speculate in the Discussion (page 13, line 406-418) about what their roles might be.

5: In line 186, the drug resistance parasites to KAE609 were emerged by day12. I think one of the reason is that the parasite with G358S acquired the strong resistance with IC50 value 400 – 600 nM. Although the parasite exhibiting weak and moderate drug resistance would emerge with same frequency as the parasite with G358S, they might become minor population in the culture due to slower multiplication in the presence of drug. As a result, it appear that only parasites with G358S may appear to have generated from mutator lines. Please discuss about this possibility.

Reply: This is an interesting point. As the selection concentration is relatively low (2.5 nM), there would be scope for parasites with moderate resistance levels to emerge, thus fitness cost may be the more likely possibility. In the recent Qiu et al study mentioned above, they examine the fitness of the G385S mutant parasite and found there was not a significant change in fitness (number of parasites produced and length of lifecycle). Thus, this mutant may outcompete less-fit lines, in addition to possessing high-level resistance. We have noted the Qiu et al findings in the Discussion (page 14, lines 435-438).

6: In line 349 - 351, the authors described the possibility about the greater mutation rate in exons than that in intergenic regions in the presence of drug. This may be examined using the transgenic parasite. For example, if the transgenic parasite with G358S shows worse growth than the parasite selected from mutator in the presence of drug, selected parasite acquired the additional mutation which assists survival. On the other hand, if the transgenic parasite with G358S shows worse growth than wild type in the absence of drug, this mutation impose the fitness cost to the parasites. This experiment may be useful for understanding the strategy of the parasite acquiring the mutation in exons.

Reply: This is a similar point to that raised by Reviewer 2, regarding the challenge of disentangling what might be functional supportive mutations rather than bystanders. In the case of the G358S mutant specifically, as mentioned above the gene-edited parasite has a similar fitness to the wild-type parent, suggesting that it may not need supporting mutations. We have however added an alternate possibility in the Discussion (page 13, line 384-387) about why in general more mutations might be observed in the presence of drug. This could also be because parasites bearing a higher burden of mutations overall may be selected against due to a fitness cost in the absence of drug, but under drug pressure a parasite that has a protective resistance mutation will outcompete all other parasites even if it is disadvantaged with a higher number of bystander mutations.

Reviewer comments, second round review

Reviewer #1 (Remarks to the Author):

The authors have addressed all the concerns that I had with the manuscript.

Reviewer #2 (Remarks to the Author):

All the points that I raised have been well revised.

Reviewer #3 (Remarks to the Author):

I think that the revised manuscript has been improved compared to the original submission version. But, I still have two comments and questions regarding the result about library screening using MMV665794 and the drug resistance mechanism by QRP1.

1: Six clones were isolated by two independent selections using MMV665794 and each of them contained one of two distinct SNVs, either G1612V or D1863Y, of QRP1. According to supplementary table 6, G1612V and D1863Y were identified from falsk 2 and 3, respectively. Please add this explanation in RESULT section. I think this information will be useful for reader to evaluate the reproducibility of screening.

2: The authors showed that G1612V and D1863Y mutations of QRP1 conferred MMV007224 resistance to the parasites. Furthermore, they suggests that disruption of QRP1 also confer similar resistance. Do those mutations disrupt the function of QRP1. In addition, is this disruption responsible for resistances against quinxaline analogues? What mechanism can be supposed? Please address those questions in DISCUSSION section.

Response to Reviewer Comments

Reviewer #3 (Remarks to the Author):

I think that the revised manuscript has been improved compared to the original submission version. But, I still have two comments and questions regarding the result about library screening using MMV665794 and the drug resistance mechanism by QRP1.

Reply: Thank you your consideration of our resubmission

1: Six clones were isolated by two independent selections using MMV665794 and each of them contained one of two distinct SNVs, either G1612V or D1863Y, of QRP1. According to supplementary table 6, G1612V and D1863Y were identified from flask 2 and 3, respectively. Please add this explanation in RESULT section. I think this information will be useful for reader to evaluate the reproducibility of screening.

Reply: We have now added that detail to the Results (line 279-281).

2: The authors showed that G1612V and D1863Y mutations of QRP1 conferred MMV007224 resistance to the parasites. Furthermore, they suggests that disruption of QRP1 also confer similar resistance. Do those mutations disrupt the function of QRP1. In addition, is this disruption responsible for resistances against quinoxaline analogues? What mechanism can be supposed? Please address those questions in DISCUSSION section.

Reply: We do not know yet whether those two SNVs result in loss of function of QRP1 without further studies to understand the biochemical function of this protein. However, we have included in the discussion this possibility, given that a frameshift mutation in QRP1 was also obtained under selection with the related analog MMV007224. We also suggest a potential mechanism similar to the PfPARE esterase – see below.

“Although we do not know whether the G1612V and D1863Y mutations confer loss of function, the presence of the frameshift mutation in the MMV007224-selected line is suggestive. Whether QRP1 acts directly on quinoxaline compounds, in a manner akin to the PfPARE esterase that converts a prodrug to an active form⁵⁰ or has an indirect effect is not known.”